# The companion of cellulose synthase 1 confers salt tolerance through a Tau-like mechanism in plants

Christopher Kesten[1,2,3], Arndt Wallmann[4], René Schneider[2,3], Heather E. McFarlane[2], Anne Diehl[4], Ghazanfar Abbas Khan[2], Barth-Jan van Rossum[4], Edwin R. Lampugnani[2], Witold G. Szymanski[3], Nils Cremer[4], Peter Schmieder[4], Kristina L. Ford[2], Florian Seiter[4], Joshua L. Heazlewood[2], Clara Sanchez-Rodriguez[1], Hartmut Oschkinat[4] & Staffan Persson[2,3]

Microtubules are filamentous structures necessary for cell division, motility and morphology, with dynamics critically regulated by microtubule-associated proteins (MAPs). Here we outline the molecular mechanism by which the MAP, COMPANION OF CELLULOSE SYNTHASE1 (CC1), controls microtubule bundling and dynamics to sustain plant growth under salt stress. CC1 contains an intrinsically disordered N-terminus that links microtubules at evenly distributed points through four conserved hydrophobic regions. By NMR and live cell analyses we reveal that two neighboring residues in the first hydrophobic binding motif are crucial for the microtubule interaction. The microtubule-binding mechanism of CC1 is reminiscent to that of the prominent neuropathology-related protein Tau, indicating evolutionary convergence of MAP functions across animal and plant cells.

[1] Department of Biology, ETH Zurich, 8092 Zurich, Switzerland. [2] School of Biosciences, University of Melbourne, Parkville, 3010 Victoria, Australia. [3] Max-Planck-Institute of Molecular Plant Physiology, Am Mühlenberg 1, 14476 Potsdam-Golm, Germany. [4] Leibniz-Forschunginstitut für Molekulare Pharmakologie (FMP), NMR-supported Structural Biology, Robert-Rössle-Str. 10, 13125 Berlin, Germany. These authors contributed equally: Christopher Kesten, Arndt Wallmann. Correspondence and requests for materials should be addressed to H.O. (email: oschkinat@fmp-berlin.de) or to S.P. (email: Staffan.persson@unimelb.edu.au)

Microtubules are tubular structures essential to morphogenesis, division and motility in eukaryotic cells[1]. While animal cells typically contain a centrosome with radiating microtubules toward the cell periphery, growing plant cells arrange their microtubules along the cell cortex[2]. A major function of the cortical microtubules in plant cells is to direct the synthesis of cellulose, a fundamental component of the cell wall essential to plant morphology[3]. Cellulose is produced at the plasma membrane by cellulose synthase (CESA) protein complexes (CSCs[4]) that display catalytically driven motility along the membrane[3]. The recently described microtubule-associated protein (MAP), COMPANION OF CELLULOSE SYNTHASE 1 (CC1), is an integral component of the CSC and sustains cellulose synthesis by promoting the formation of a stress-tolerant microtubule array during salt stress[5]. As cellulose synthesis is key for plant growth, engineering of plants to better produce cellulose is of utmost importance to agriculture. Indeed, understanding the molecular mechanism by which CC1 controls cellulose synthesis may bear opportunities to improve cultivation on salt-affected lands.

The microtubule network is highly dynamic, and its state is influenced by the action of MAPs. The mammalian Tau/MAP2/MAP4 family represents the most investigated MAP set, primarily due to Tau's importance in the pathology of neurodegenerative diseases[6–8]. In vitro, Tau promotes polymerization and bundling of microtubules, and diffuses along the microtubule lattice[9–11]. In the brain, Tau is predominantly located at the axons of neurons, where it contributes to the microtubule organization that drives neurite outgrowth[12,13]. In disease, Tau self-aggregates into neurofibrillary tangles that might trigger neurodegeneration[14]. Intriguingly, no clear homologs of the Tau/MAP2/MAP4 family have been identified in plants[15,16]. Because, the full scope of Tau's biological role remains elusive, identification of Tau-related proteins outside the animal Kingdom would benefit our understanding of how this class of MAPs functions.

In this study, we unravel the microtubule-binding mechanism of CC1 and show that it is reminiscent to that of Tau, indicating evolutionary convergence of MAP functions across animal and plant cells.

## Results

**The N-terminus of CC1 bundles microtubules.** The cytosolic N-terminal part of CC1 (residues 1–120, CC1ΔC223) binds to microtubules and restores microtubule reassembly, cellulose synthesis and wild-type growth of *cc1cc2* (null-mutation in CC1 and its closest homolog CC2) seedlings on high levels of salt[5]. These data indicate that CC1ΔC223 is critical to CC1's function during stress, and we therefore set out to investigate the molecular details of how it interacts with microtubules. We cross-linked 6xHis-tagged CC1ΔC223 with α-β-tubulin dimers using 1-ethyl-3-(3-dimethylaminopropyl) carbodiimide hydrochloride (EDC)[17], which led to di- and multimeric protein products (Fig. 1a, Supplementary Fig. 1a). We used EDC, which links functional groups of lysine to either aspartate or glutamate, and not the typical sulfhydryl-reactive or lysine–lysine cross-linkers, as CC1ΔC223 only contains a single cysteine and has a basic isoelectric point (pI) (i.e. the main reactive amino acid is lysine), while tubulin/microtubules have an acidic pI (i.e. the main reactive amino acids on the surface of the tubulin dimer are aspartate and glutamate). After LC/MS/MS analysis, we used four different software packages (StavroX, pLink, SIM-XL, Crux[18–21]) to identify potential inter-cross-links between tubulin and CC1ΔC223. Extensive manual curation resulted in five well-defined covalent bonds between CC1ΔC223 and α- or β-tubulin (Fig. 1b). We consistently detected four peptides of CC1ΔC223

cross-linked to β-tubulin ($K^{40}$–$E^{111}$, $K^{94}$–$E^{111}$, $K^{96}$–$E^{111}$ and $K^{96}$–$E^{158}$; letters and numbers indicate amino acids in CC1ΔC223 and β-tubulin, respectively; Fig. 1b, c; Supplementary Table 2 and Supplementary Fig. 1b-h). Notably, the three sequentially distant $K^{40}$ and $K^{94/96}$ of CC1ΔC223 cross-linked to the same residue on β-tubulin ($E^{111}$). This suggests that two CC1 regions might bind the same sites on two different β-tubulin molecules, which is corroborated by the multimeric protein products in the SDS page. The cross-linked position on α-tubulin is close to the hydrophobic interface between tubulin heterodimers, a site that is frequently occupied by agents that directly regulate microtubule formation, such as vinblastine, the stathmin-like domain (SLD) of RB3, and also by Tau[22–24] (Supplementary Fig. 2a-b).

To further investigate how CC1ΔC223 binds microtubules, we co-polymerized tubulin in the presence of CC1ΔC223. We then labeled CC1ΔC223 using 5 nm gold-conjugates that recognize the His-tag[25] and monitored the formed microtubules and gold distribution via transmission electron microscopy (TEM). Gold labeling only occurred at closely aligned microtubules with very small inter-microtubule distances (Fig. 1d, e, Supplementary Fig. 1d-e), and were visible as evenly distributed foci in straight rows along interphases of two neighboring microtubules (Fig. 1d, e), while a negative control employing BSA did not show any specific microtubule labeling (Supplementary Fig. 2c). The gold particles were typically spaced by 10 nm (Fig. 1f; 10.0 nm ± 2.4 nm; mean ± S.D.; three-independent experiments; $n = 1785$ labels). The number of gold-labels in a given row ranged between two and 41 labels (Fig. 1g; 8 ± 5 labels; mean ± S.D; three independent replicates; $n = 274$ rows), making each row about 80 nm in length. We also observed multiple gold-labeled rows on one microtubule when in close proximity to several other microtubules (Supplementary Fig. 2d). The angles between gold-labeled rows were small (Fig. 1h; 2.8° ± 3°; mean ± S.D.; three-independent replicates; $n = 98$ rows), highlighting that the labeling did not shift between neighboring protofilaments on the same microtubule. These data indicate that CC1ΔC223 promotes microtubule bundling. Indeed, increasing levels of CC1ΔC223 correlated with increased microtubule bundling in TEM experiments (Fig. 2a, b), while a BSA control did not show increased bundling (Supplementary Fig. 3a, b).

**The CC1 N-terminus can diffuse along the microtubule lattice.** As our TEM experiments only provide static information on the interactions between CC1ΔC223 and microtubules, we labelled the sole sulfhydryl group ($C^{116}$) in CC1ΔC223 with the green fluorescent dye CF488A-maleimide (Supplementary Fig. 3c) and performed rhodamine-labeled microtubule interaction assays[26]. Using total internal reflection fluorescence microscopy, we observed most of the CF488A-labeled CC1ΔC223 proteins as fluorescent foci associated with microtubules (Fig. 2c, Supplementary Movie 1). CF488A-labeled CC1ΔC223 diffused bidirectionally along the rhodamine-labeled microtubules and occurred on both single and bundled microtubules (Fig. 2d). In accordance with the results above, CF488A-labeled foci occupied bundled microtubules for a longer time than single microtubules (Fig. 2e, Supplementary Fig. 3d-e). The diffusion coefficient (0.076 ± 0.007 μm² s⁻¹; mean ± S.D.; $n = 50$ molecules) of fluorescent foci exhibited a linear relationship with time (Supplementary Fig. 3f), indicating free diffusion. These data are reminiscent to that of Tau, which promotes microtubule-bundling and polymerization, and also moves along microtubules in vitro with comparable diffusion coefficients (0.142–0.292 μm² s⁻¹[10]).

**The N-terminus of CC1 is intrinsically unstructured.** To understand how CC1ΔC223 engages with microtubules, we

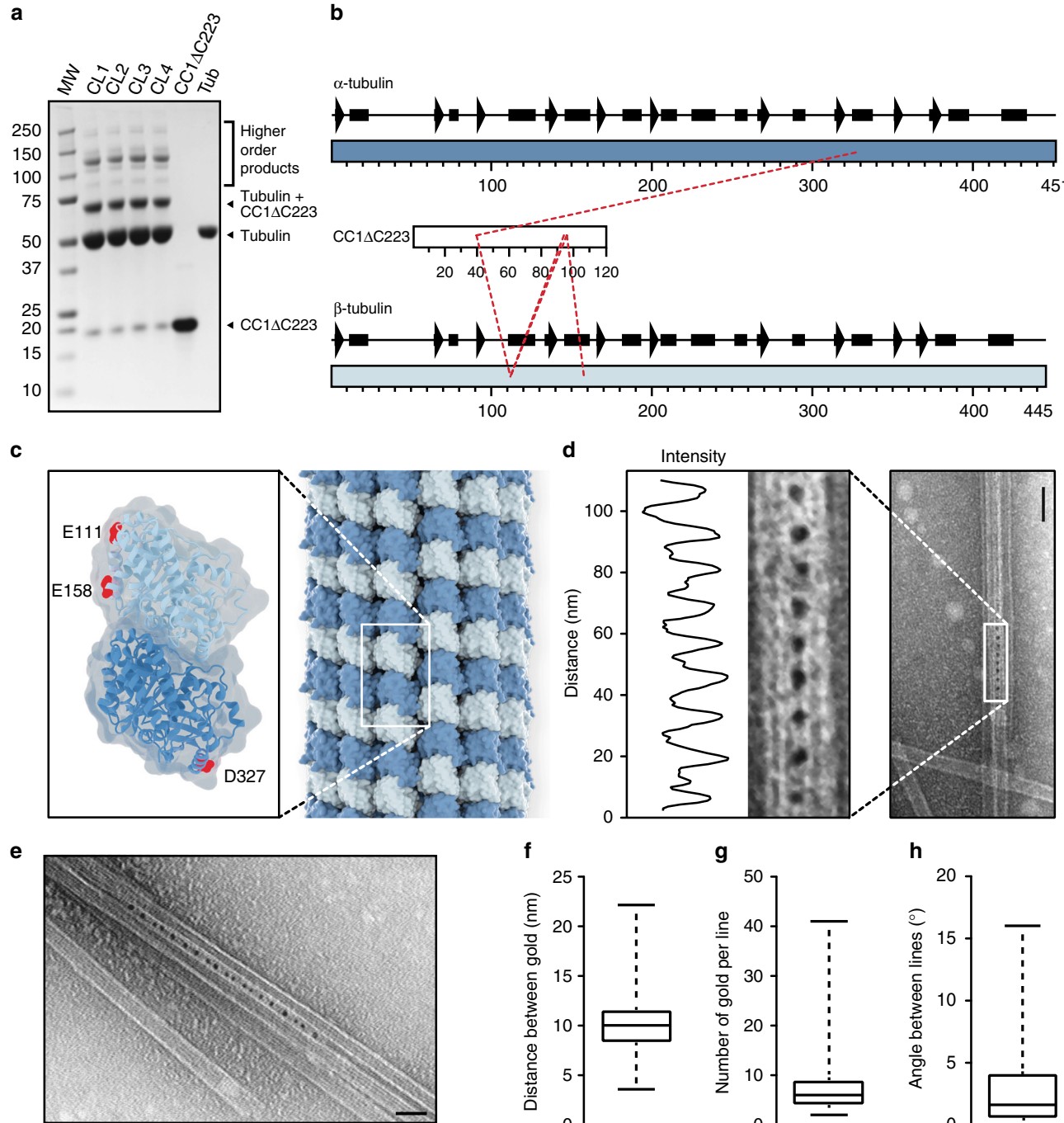

**Fig. 1** The N-terminus of CC1 binds sites on both α- and β-Tubulin and cross-links microtubules. **a** SDS-Page of EDC-induced cross-linking of 6xHis-CC1ΔC223 (16 kDa) and tubulin dimers (2 × 55 kDa). Arrowheads depict position of relevant protein bands. MW molecular weight marker, CL1-4 cross-linking reaction 1–4. Higher order cross-linking products represent cross-links between e.g. tubulin +2 × CC1ΔC223 (87 kDa), tubulin dimers (110 kDa), tubulin dimers + CC1ΔC223 (126 kDa), tubulin dimers +2 × CC1ΔC223 (142 kDa). **b** Schematic views of the secondary structures of α- and β-tubulin, and the CC1ΔC223 sequence. Dashed lines depict detected cross-linking positions of CC1ΔC223 and α- or β-tubulin. **c** Projection of detected cross-links onto an α/β-tubulin dimer (PDB code 1tub). Dark blue = α-tubulin; Light blue = β-tubulin; Sites for cross-linked amino acids are marked in red. **d** Representative TEM image of CC1ΔC223 distribution along negatively stained, taxol-stabilized microtubules polymerized in the presence of 6xHis-CC1ΔC223. CC1ΔC223 protein is visualized by a 5 nm gold-conjugated Ni-NTA tag that recognizes 6xHis-tagged proteins. A transect was taken along rows of gold particles, and dips in the light intensity along the transect correspond to gold particle centers. Note the even distribution of the electron-dense gold particles in between neighboring microtubules. Scale bar = 50 nm. **e** CC1ΔC223 distribution along negatively stained, taxol-stabilized microtubules polymerized in the presence of 6xHis-CC1ΔC223. CC1ΔC223 can form a zipper-like pattern that links microtubules. Scale bar = 100 nm. **f** Quantification of the distance between individual gold particles as shown in **d** and **e** (box plot: Center lines show the medians; box limits indicate the 25th and 75th percentiles; whiskers extend to the minimum and maximum). **g**, **h** Quantification of number of gold labels per row (**g**) and the angle between adjacent gold-labeled rows (**h**) from images as those in **d** and **e** (box plots: Center lines show the medians; box limits indicate the 25th and 75th percentiles; whiskers extend to the minimum and maximum)

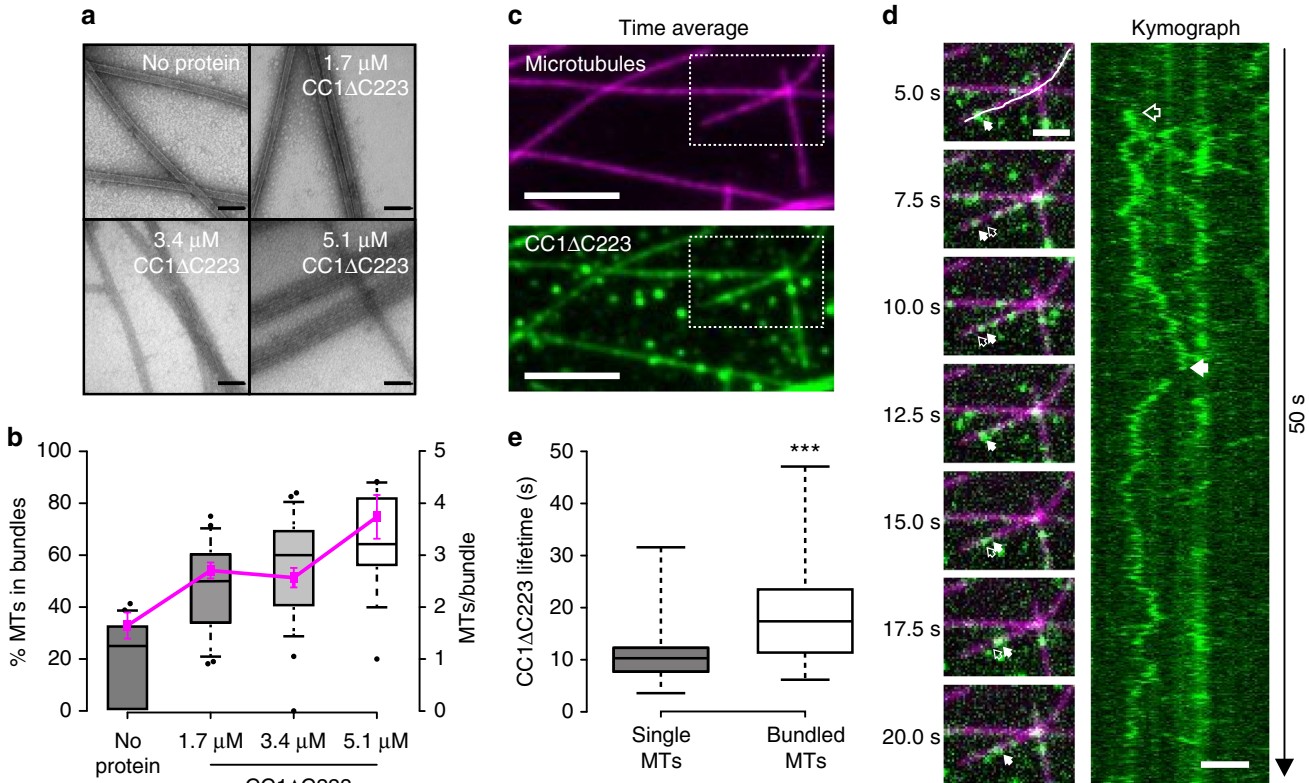

**Fig. 2** The N-terminus of CC1 induces microtubule bundling and can diffuse along the microtubule lattice. **a** Transmission electron microscopy (TEM) of negatively stained taxol-stabilized microtubules after addition of increasing levels of 6xHis-CC1ΔC223 during microtubule polymerization. Note that it is very difficult to discern individual microtubules in the microtubule bundles after addition of ~3 μM of CC1ΔC223. Scale bars = 100 nm. **b** Quantification of the proportion of microtubules in bundles (left y-axis, box plots: Center lines show the medians; box limits indicate the 25th and 75th percentiles; whiskers extend to the 10th and 90th percentiles, outliers are represented by dots) and number of microtubules/bundle (right y-axis, magenta line: mean ± SEM) with increasing concentration of 6xHis-CC1ΔC223 (quantified from images such as those in **a**). **c** CF488A-labeled 6xHis-CC1ΔC223 proteins (green) associated with surface-bound microtubules (magenta) in vitro. Scale bar = 5 μm. **d** Time-series images (left panel) of CF488-labeled 6xHis-CC1ΔC223 (green) diffusing along microtubules (magenta). Filled arrow = position in current frame, empty arrow = position in previous frame. Scale bar = 2 μm. Representative kymograph (right panel) along solid line in left panel (top) showing diffusion of 6xHis-CC1ΔC223 foci. Scale bar = 2 μm. **e** 6xHis-CC1ΔC223 lifetime on single versus bundled microtubules (box plots: Center lines show the medians; box limits indicate the 25th and 75th percentiles; whiskers extend to the minimum and maximum), n = 60 single and 37 bundled microtubules, ***p-value < 0.001, Welch's unpaired t-test)

assessed its structural features using "solution state" NMR, circular dichroism spectroscopy (CD) and analytical ultracentrifugation (AUC). The 2D $^1$H–$^{15}$N-heteronuclear single quantum coherence (HSQC) spectrum of $^{15}$N-labeled CC1ΔC223 showed narrow signals and poor chemical shift dispersion in the $^1$H dimension, which is characteristic for intrinsically disordered proteins (Fig. 3a). For the sequence-specific assignment, we used a combination of three-dimensional and four-dimensional experiments with non-uniform sampling to assign ~85% of the backbone resonances. The disordered nature of CC1ΔC223 was supported via multiple sequence data analysis algorithms and CD measurements (Supplementary Fig. 4a, Fig. 3b). AUC analysis revealed only elongated monomeric forms of the protein in solution (Fig. 3c). To estimate local propensities for secondary structure formation, we determined neighbor-corrected structural propensities using the ncSPC algorithm[27]. Experimental Cα, Cβ and C' chemical shifts were subtracted from the reference random-coil state of the IDP-based ncIDP chemical shift library (Supplementary Fig. 4b–d)[28]. The resulting propensity score revealed few and rather scattered deviations from random coil values (Fig. 3d). Moreover, the uniform and fast dynamics of CC1ΔC223 are consistent with a disordered, highly dynamic and monomeric state in solution (Supplementary Fig. 4e–g), similar to the members of the Tau/MAP2/MAP4 family.

**CC1 engages with microtubules via four hydrophobic motifs.** To study CC1ΔC223-microtubule interactions in a residue-specific manner, we recorded $^1$H–$^{15}$N HSQC spectra of $^{15}$N-labeled CC1ΔC223 in the presence and absence of taxol-stabilized microtubules. We observed line broadening and vanishing of individual cross-peaks when microtubules were added (Fig. 4a, b). The effects of the microtubules on the transverse relaxation rate ($\Delta R_2$) of CC1ΔC223 signals were reversible, residue specific, independent of the magnetic field, did not correlate with the chemical shift changes, and relaxation dispersion experiments did not show contributions of intermediate exchange (Supplementary Fig. 5a–g). To conclude, the line broadening is a direct result of CC1ΔC223-microtubule complex formation. Figure 4c shows the intensity ratios of cross-peaks taken from 3D HNCA spectra of $^{15}$N,$^{13}$C-labeled CC1ΔC223 in the presence and the absence of microtubules ($I_{bound}/I_{free}$) per residue. A significant intensity decrease is observed in four regions, comprising residues $^{23}$RPVYYVQS$^{30}$, $^{45}$FHSTPVLSPM$^{54}$, $^{74}$FSGSLKPG$^{83}$ and $^{103}$QWKECAVI$^{110}$ (Fig. 4c). Due to signal overlap, the region between residues 60 and 80 is not well covered. We found a clear correlation between the NMR-based microtubule-interaction profile and the hydrophobicity pattern of CC1ΔC223, highlighting the role of hydrophobic interactions (Fig. 4d). The binding motifs are separated by stretches of mobile residues, presumably acting as linkers that are likely to retain a high degree of

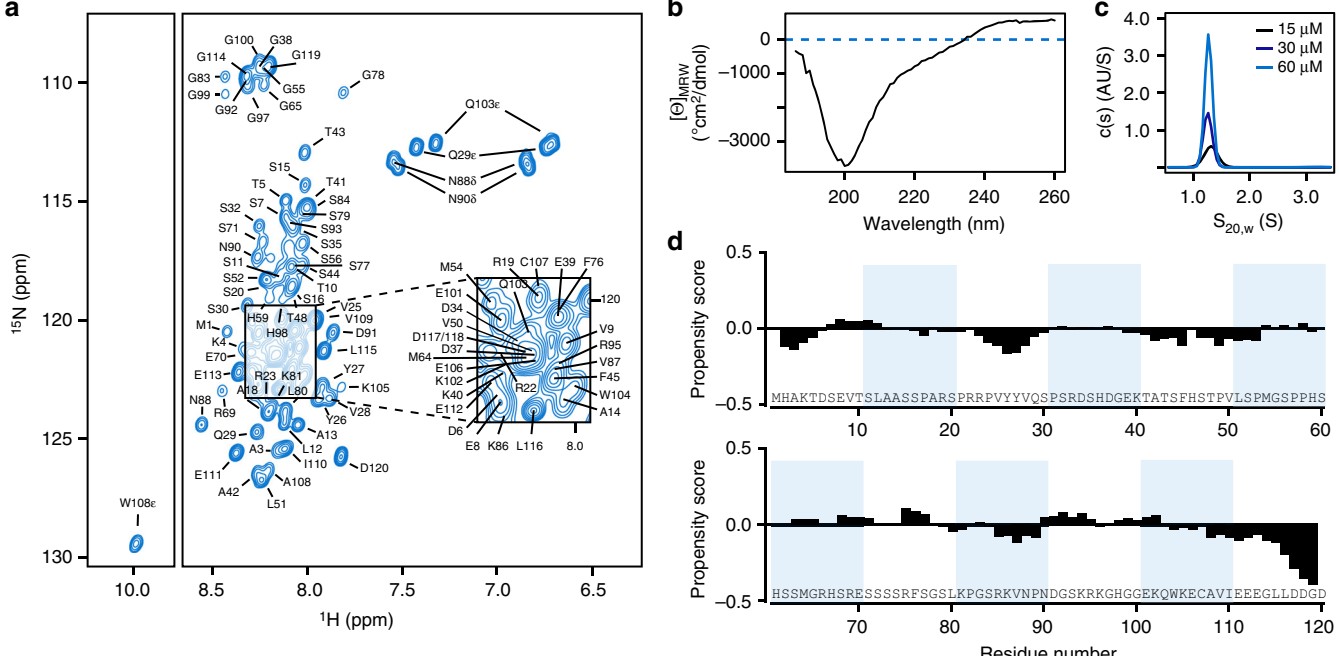

**Fig. 3** Structural characteristics of the CC1 N-terminus. **a** Assigned $^1$H–$^{15}$N HSQC spectrum of $^{15}$N-labeled CC1ΔC223 in solution. The low signal dispersion in the $^1$H dimension is characteristic of an intrinsically disordered protein. **b** Circular dichroism (CD) spectrum of 6xHis-CC1ΔC223 in solution supporting lack of clear structures in the protein. **c** Analytical ultracentrifugation at three different CC1ΔC223 concentrations showed a single size population at the approximate molecular weight of monomeric CC1ΔC223. The frictional coefficient of 1.7 is characteristic for elongated protein shapes. **d** Structural propensity plot of CC1ΔC223 calculated by ncSPC[27] using ΔδCα, ΔδCβ and ΔδC′ secondary chemical shifts (Supplementary Fig. 4b–d). Propensity scores of 1 and −1 report on fully formed α- or β-structure, respectively

flexibility thus facilitating a highly dynamic interaction with microtubules. This binding behavior is reminiscent to that of Tau[29], and the microtubule-binding regions of CC1ΔC223 also share remarkable similarities in hydrophobicity, size, and spacing with those of the microtubule-binding regions of Tau(201–320) (Fig. 4e).

**Two tyrosine residues contribute to CC1 microtubule binding**. Microtubule binding of the four regions individually was investigated by saturation transfer difference (STD) NMR measurements (Supplementary Fig. 6a). The peptides CC1(16–38), CC1(41–64), CC1(65–85) and a positive control peptide Tau (211–242) yielded strong STD intensities in the amide and aromatic regions of the $^1$H spectrum (Supplementary Fig. 6b–e). No significant STD effects were observed for a negative control peptide, CC1(83–103), corresponding to the third poorly conserved linker region, and for the most C-terminal region CC1 (100–114) (Supplementary Fig. 6f–g). Targeting the N-terminal binding site, the exchange of $^{26}$YY$^{27}$ to alanine in a CC1YYAA (16–38) peptide resulted in a substantially reduced STD profile, corroborating a contribution of these aromatic rings to the interaction (Supplementary Fig. 6h). Indeed, the same mutation in CC1ΔC223 resulted in significantly reduced signal broadening of residues in the N-terminal region, while the intensity ratios for the C-terminal part remained similar to the wild-type protein (Fig. 4f). Likewise, the mutated CC1ΔC223 bound to microtubules with a lower affinity compared to the wild-type sequence in microtubule spin down assays (Supplementary Fig. 6i–j), corroborating an important function of the two tyrosine residues in microtubule binding.

**Mutation of CC1 impairs CESA movement**. To assess how mutations in the two microtubule-binding tyrosine residues affect the function of CC1 in vivo, we mutated them to alanine in the full-length CC1 (CC1YYAA), fused it N-terminally with GFP, and transformed it into *Arabidopsis thaliana cc1cc2* mutant plants. The *cc1cc2* mutant seedlings display reduced growth and crystalline cellulose content on salt-containing media[5]. These phenotypes were not restored in *cc1cc2* GFP-CC1YYAA seedlings when grown on salt-containing media as compared to controls (Fig. 5a–c).

Spinning-disc confocal microscopy showed GFP-CC1YYAA signals as distinct foci at the plasma membrane (Supplementary Movie 2) and within cytoplasmic compartments in dark-grown Arabidopsis hypocotyl cells, in accordance with reports on GFP-CC1[5] (Supplementary Fig. 7a-c). GFP-CC1 co-localizes and migrates with tdTomato(tdT)-CESA6, which is an important subunit of the CSC[30], at the plasma membrane[5]. Notably, the GFP-CC1YYAA also co-migrated with tdT-CESA6 at the plasma membrane on MS media without addition of salt (Supplementary Fig. 7a-b, d-e; Pearson correlation coefficient $r = 0.74 \pm 0.06$; mean ± S.D, six cells from six seedlings and three-independent experiments). However, in contrast to GFP-CC1, the migration of GFP-CC1YYAA was largely independent of cortical microtubules (mCherry (mCh)-TUA5[31]; Fig. 5d–f). This indicates that reduced microtubule binding of GFP-CC1YYAA either directly affects the ability of CSCs to engage with microtubules, or that the microtubule array is mis-regulated and cannot fulfill its guiding function anymore.

**Mutation of CC1 disrupts salt tolerance**. To investigate whether the CC1YYAA can sustain microtubule and CSC function during salt exposure, we exposed seedlings to 200 mM salt and recorded time series of microtubule (mCh-TUA5) and CC1 (GFP-CC1 or GFP-CC1YYAA) behavior (Supplementary Fig. 7f). The GFP-CC proteins (either GFP-CC1 or GFP-CC1YYAA) were considered as proxy for the CSC behavior because they co-localize and migrate together with tdT-CESA6. In agreement with[5], the

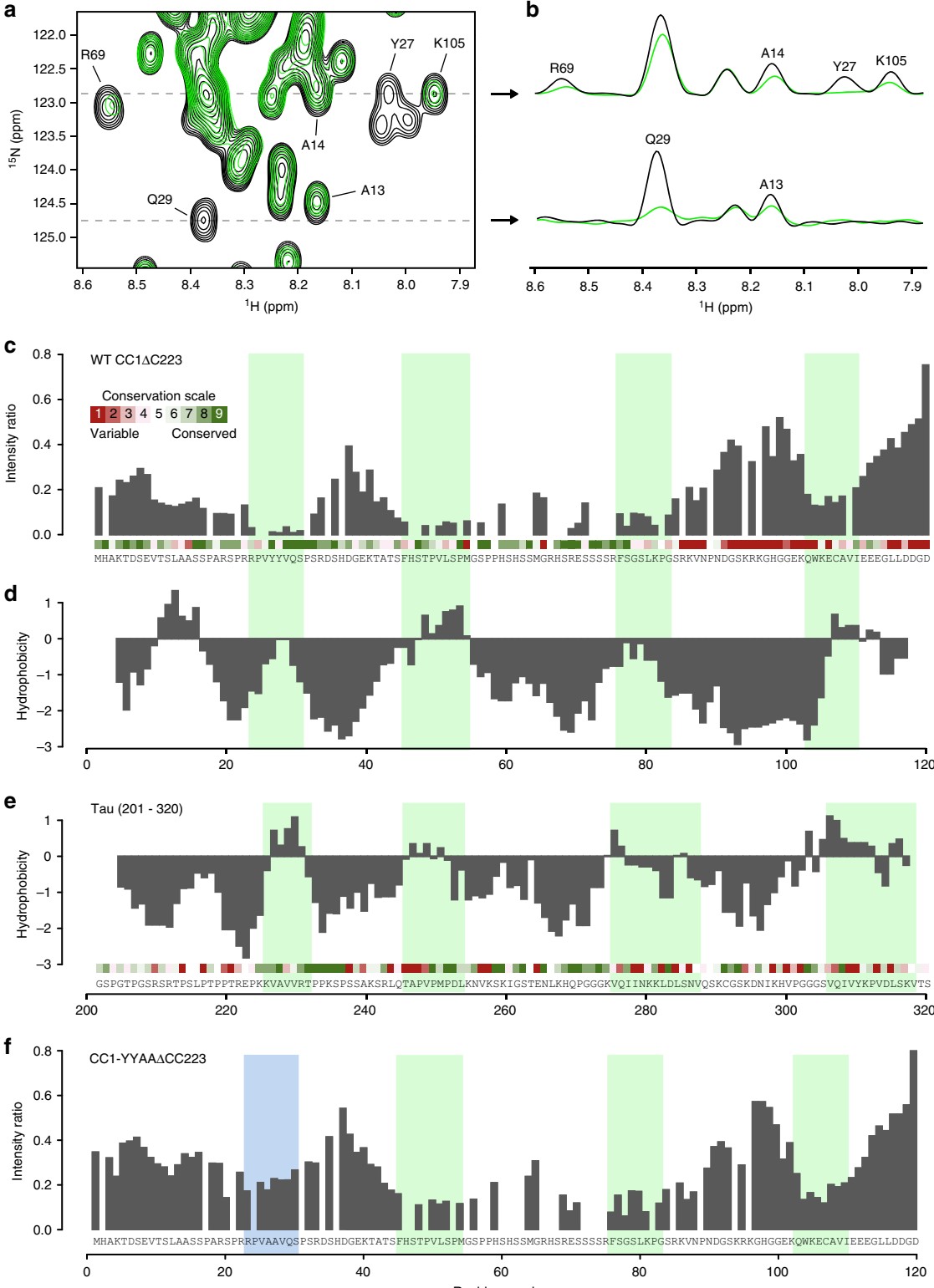

**Fig. 4** The N-terminus of CC1 binds to taxol-stabilized microtubules via short hydrophobic and conserved regions. **a** $^1$H–$^{15}$N HSQC spectrum of free CC1ΔC223 (black) and in the presence of equimolar taxol-stabilized microtubules (green). Selected residues are labeled. **b** F$_2$-cross sections, showing $^1$H-signals, taken along dotted lines in **a** at $^{15}$N frequencies 122.9 and 124.7 ppm. **c** Intensity ratio of free CC1ΔC223 HNCA signals and in complex with microtubules. Minima are highlighted with green bars. Site-specific evolutionary conservation calculated by CONSURF is plotted above the sequence in a color code (green = conserved, red = unconserved). **d** Hydrophobicity scores of CC1ΔC223 according to the Kyte-Doolittle scale, calculated in a 5-residue window. **e** Hydrophobicity scores of Tau(201–320) according to the Kyte-Doolittle scale, calculated in a 5-residue window. Sequence conservation is plotted above the sequence like in **c**. Green bars highlight the interacting regions of Tau with microtubules as in ref. [29]. **f** Intensity ratio of free CC1YYAAΔC223 HNCA signals and in complex with microtubules. Mutated N-terminal region highlighted with blue bar

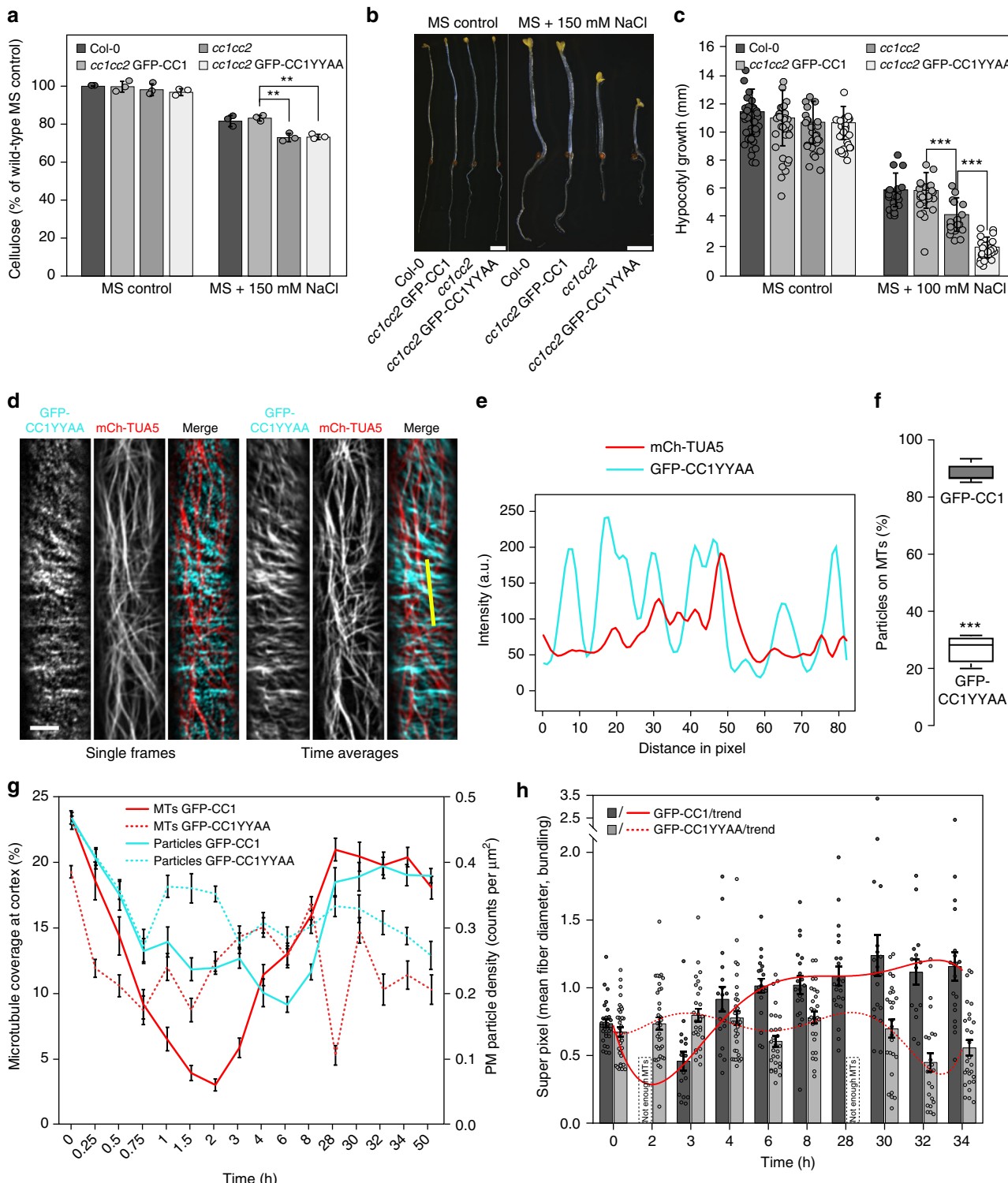

microtubule array and cellulose synthesis were restored within 28 h of salt exposure in the GFP-CC1-complemented cc1cc2 seedlings (Fig. 5g). However, the cc1cc2 GFP-CC1YYAA-complemented seedlings largely mimicked the cc1cc2 mutant seedlings and failed to restore the microtubule array and cellulose synthesis during the course of the experiment (Fig. 5g). Interestingly, while the cc1cc2 GFP-CC1 line showed increased microtubule bundling of the salt-adjusted microtubule array, the cc1cc2 GFP-CC1YYAA cells failed to do so (Fig. 5h). Furthermore, the microtubule dynamics differed in the GFP-CC1 and

GFP-CC1YYAA cell lines (Supplementary Fig. 7g-h), indicating that the microtubule dynamics and bundling are key to build a salt-tolerant microtubule array. Hence, the YY-containing region of CC1 is necessary to sustain microtubule array organization and cellulose synthesis during salt stress.

## Discussion

Abiotic stress, such as soil salinity, substantially impacts plant growth[32] and thus dramatically compromises global agricultural

**Fig. 5** Mutations in the first microtubule-binding region of CC1 impair salt tolerance of plants because of mis-regulated microtubule organization upon salt stress. **a** Cellulose levels in seedlings grown as in **b**. Values are means ± S.D. expressed as % cellulose of wild-type seedlings grown on MS control media. Dots represent individual data points of the corresponding bars. $N \geq 3$ biological replicates with two technical replicates each from three-independent experiments. Welch's unpaired $t$-test; **$p$-value $\leq 0.01$. **b** Seedlings germinated and grown for 2 days on MS plates and then transferred to either MS control plates or MS plates supplemented with 150 mM NaCl and grown for additional 5 days. Scale bar = 2 mm. Please be aware that the images were stitched with Leica LAS X Life Science software. **c** Quantification of hypocotyl elongation of seedlings grown on MS plates for 3 days and then transferred to either MS control plates or MS plates supplemented with 100 mM NaCl and grown for additional 4 days. Dots represent individual data points of the corresponding bars. Values are mean ± S.D., $n = 30$ seedlings, 10 seedlings each per three-independent experiments. Welch's unpaired $t$-test; ***$p$-value $\leq 0.001$. **d** GFP-CC1YYAA and mCh-TUA5 in dual-labeled 3-day-old $cc1cc2$ etiolated hypocotyls (left panels; single frame, right panels; time average projections). Scale bars = 5 μm. **e** Fluorescence intensity plot of GFP-CC1YYAA and tdT-CESA6 from transect in **d** along the depicted yellow line. Note that the GFP signal does not substantially correlate with the mCherry signal. **f** Quantification of GFP-CC1 and GFP-CC1YYAA fluorescent foci on cortical microtubules in a $50 \times 50$ pixel area of five individual time-lapse images, $n = 5$ cells from 5 seedling and three independent experiments (box plots: Center lines show the medians; box limits indicate the 25th and 75th percentiles; whiskers extend to the minimum and maximum). Welch's unpaired $t$-test; ***$p$-value $\leq 0.001$. **g** Quantification of microtubule and GFP-CC (GFP-CC1 or GFP-CC1YYAA) coverage at the cell cortex and plasma membrane, respectively, after exposure of $cc1cc2$ seedlings to 200 mM NaCl as in an experiment shown in Supplementary Fig. 7f. Time indicates time after salt exposure. Values are mean ± S.E.M., $n = 27$ cells from three seedlings per time point and three-independent experiments. Two-way ANOVA analysis of microtubule coverage; $p \leq 0.001$ (genotype), $p \leq 0.001$ (time), $p \leq 0.001$ (genotype × time). Two-way ANOVA analysis of GFP-CC protein density; $p \leq 0.01$ (genotype), $p \leq 0.001$ (time), $p \leq 0.001$ (genotype × time). **h** Quantification of microtubule bundling after exposure of $cc1cc2$ GFP-CC1 /GFP-CC1YYAA seedlings to 200 mM NaCl as in an experiment shown in Supplementary Fig. 7f. The salt-adjusted microtubule array in GFPCC1 seedlings shows increased bundling after exposure to salt while the array GFPCC1YYAA seedlings does not. Dots represent individual data points of the corresponding bars. Values are mean ± S.E.M., $n = 27$ cells from three seedlings per time point and three-independent experiments. Two-way ANOVA analysis of microtubule bundling (excluding T2 and T28); $p \leq 0.001$ (genotype), $p \leq 0.001$ (time), $p \leq 0.001$ (genotype × time)

---

productivity (~50–80% loss in yield[33,34]). Unraveling molecular mechanisms that can be used to engineer plants for better stress tolerance is therefore of urgent importance. A potential target is the CC protein family that enables plants to sustain cellulose synthesis and the integrity of the cortical microtubule array during salt exposure[5]. Here, we describe how CC1 mediates the formation of a stress-stable cortical microtubule array. CC1 contains four motifs that transiently engage with microtubules and that enable microtubule polymerization and bundling, which facilitate microtubule reassembly after stress. The hydrophobic interactions of the CC1-microtubule complex could permit a more robust binding under conditions of high-ionic strength, corroborating the importance of the protein's function during salt stress. The two tyrosine residues in the most N-terminal microtubule-binding region of CC1 are key to the microtubule binding, both in vitro and in vivo. Mutations in these residues disrupted microtubule-guided CSC movement and led to failure in the generation of a stress-tolerant microtubule array.

Our results show that the microtubule binding characteristics of CC1ΔC223 are remarkably similar to that of Tau, while there is a characteristically different overall sequence architecture (Fig. 6a). Both Tau and CC1ΔC223 are intrinsically disordered proteins that can diffuse bidirectionally along the microtubule lattice[10,29]. While the typical PGGG-containing repeats of the Tau microtubule-binding domain (R1–R4) are not obvious from the CC1 sequence, the two proteins do contain four similarly spaced hydrophobic microtubule-binding regions (regions 1–4 in Fig. 6a, top). A sequence comparison of these four regions (Fig. 6a, bottom) reveals a surprisingly high number of identical or similar residues, implying evolutionary convergence of the microtubule-binding mechanism. A Tau fragment encompassing the four NMR-derived microtubule-binding regions (Tau (208–324); TauF4) joins microtubules wall-to-wall similar to that of CC1ΔC223[35]. In-depth NMR studies using TauF4[36], and cryo-EM studies on full-length Tau[8], proposed that microtubule-bound Tau spans multiple tubulin heterodimers along the microtubule principal axis. The equivalence of cross-linked positions on α-tubulin[22] between CC1ΔC223 and Tau and the longitudinal microtubule decoration of CC1ΔC223 in the gold-labeling experiments could suggest a similar interaction of CC1ΔC223 with microtubules. Comparable to the effects of

tyrosine to alanine mutations in CC1, disease-related mutations in Tau cause distinct defects in microtubule organization[37]. Furthermore, Tau-depleted rat neurons exhibit a reduction of microtubule dynamics[38], similar to what we observed for the mis-regulated microtubule array in both the $cc1cc2$ knockout[5] and the CC1-YYAA complemented mutant. Further functional and structural analogies between Tau and CC1 are reflected in the fact that both Tau and CC1 are relevant for the organism to function during stress conditions; CC1 promotes cellulose synthesis during salt stress[5]; whereas, Tau has emerged as a key regulator of stress-induced brain pathology in mice and oxidative stress in cultured fibroblasts[39,40].

While there are functional and structural analogies between CC1 and Tau, other features of the two proteins are clearly different. For example, CC1 contains a putative transmembrane and an apoplastic domain[5], whereas Tau is a cytoplasmic protein with an N-terminal projection domain that regulates microtubule spacing (Fig. 6a)[41]. Moreover, CC1 is a core component of the CSC, which is primarily localized on bundled cortical microtubules and its movement is guided by cortical microtubules in plant interphase cells (Fig. 6b)[3]. In this setting, the CC1 microtubule-binding regions interact with tubulin dimers in one, or multiple microtubules, and the microtubule-binding motif that contains the two tyrosine residues essential for stress-stable microtubule array formation is most distal to the plasma membrane. Given the local environment of CC1, i.e. being part of the CSC and integral to the plasma membrane, this distal motif might be the most prominently exposed of the four microtubule-binding motifs and therefore also most prominent in the microtubule engagement. Notably, the microtubule arrays have different design principles in animal and plant cells. The centrosome-coordinated microtubules in animal cells typically radiate from the cell center towards the periphery, while growing plant cells have a cortical microtubule array, with evenly distributed microtubules along the cell cortex[2]. We speculate that the differences in the protein domain architecture and localization of Tau and CC1 coincide with this differential microtubule array organization. In this context, the CC1 proteins are superbly situated to modulate microtubule dynamics and bundling to optimize cellulose synthesis under different environmental conditions (Fig. 6b). Engineering the microtubule-binding properties

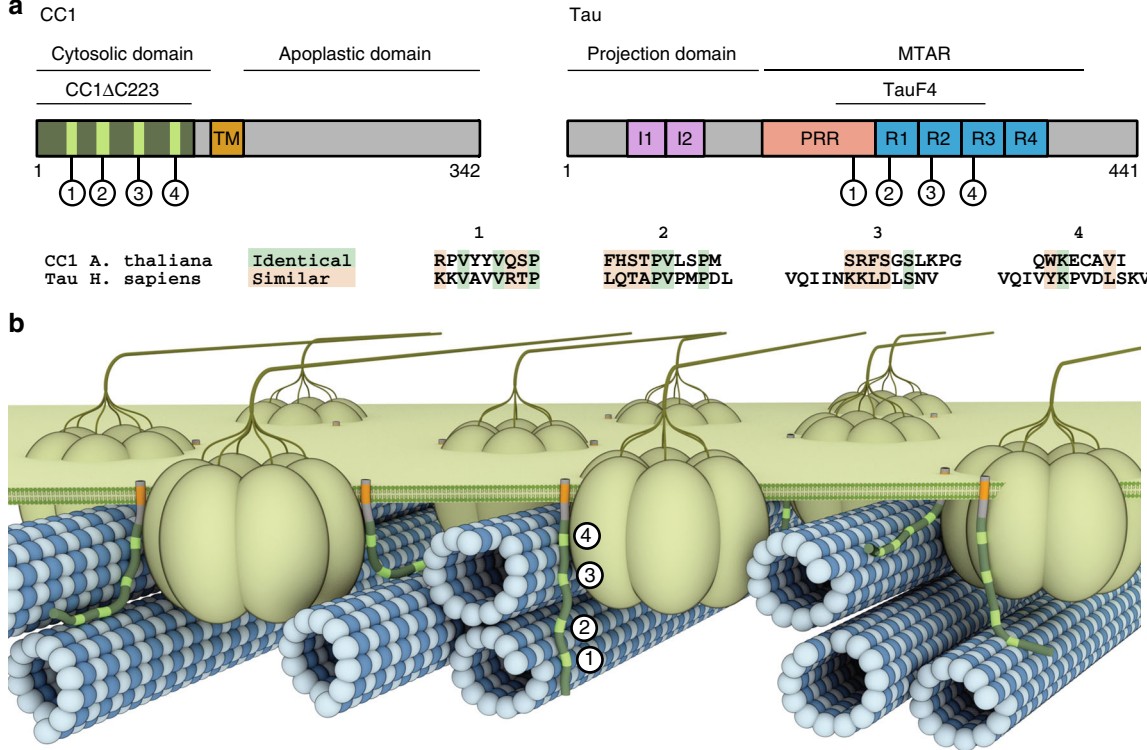

**Fig. 6** Cartoon overview of the CC1-microtubule interaction and its similarity to Tau despite sequential differences. **a** Cartoon representation of the CC1 and hTau40 domain architecture. CC1 is composed of a cytosolic N-Terminus, transmembrane domain (TM) and an apoplastic C-terminal domain. hTau40 represents the longest isoform in humans containing two N-terminal inserts (I1, I2) in the projection domain, a polyproline region (PRR) and four imperfect repeats (R1–R4) in the microtubule assembly region (MTAR). The position of the four respective microtubule-binding regions are marked and highlighted green in CC1. The amino acid sequences below compare the NMR-derived microtubule-binding regions of CC1 and Tau. Identical (green) and similar (orange; score of ≥0 in the BLOSUM62 matrix) amino acids are highlighted. **b** CC1 localization in its cellular context as part of the cellulose synthase complex (CSC). While the CSCs migrate along cortical microtubules during cellulose production, CC1 is involved in microtubule array organization via its cytosolic N-terminus. Similar to the Tau binding behavior, the CC1-microtubule interaction is dynamic and the microtubule-binding motifs (green) are potentially able to bind several tubulin dimers, distributed over one or several microtubules, thereby affecting microtubule bundling or dynamics. The individual components of the cartoon representation are drawn to scale, except for CC1 thickness and omission of its apoplastic domain. Membrane association and length of CC1, as well as the microtubule diameter, determine possible interaction modes

of this domain, perhaps by design principles of Tau, might improve cellulose synthesis and thus biomass production on salt-affected lands.

## Methods

**Plant material and growth**. T-DNA insertion lines cc1 (SAIL_838_F07) and cc2 (GK_511F01) and double mutant lines cc1cc2 were described earlier[5]. Primers used to screen for homozygous insertion lines can be found in Supplementary Table 1 (primers 1–7).

For genotyping the cc1 mutant, cc1 LP + cc1 RP were combined for wild-type PCRs, for insertion PCRs cc1 RP + LB_SAIL were combined. For genotyping the cc2 mutant, cc2 LP + cc2 RP were combined for wild-type PCRs, for insertion PCRs cc2 insert fw + 08409_GABI Insert were combined. To check for the insertion of the YY to AA mutated version of GFP-CC1, plants were genotyped with primers listed in Supplementary Table 1 (primers 8–9).

The marker lines mCherry-TUA5[31] and tdTomato-CesA6[30] have been described previously. Arabidopsis plants were grown as described[30,42]. The lines cc1cc2 GFP-CC1 and cc1cc2 GFP-CC1ΔN120 were described previously[5]. Seedlings were grown on control (MS media + 1% sucrose) and plates supplemented with 2 nM isoxaben or 300 nM oryzalin and hypocotyl elongation was quantified after 4 and 7 days of growth, respectively. For salt treatments, seeds germinated and grown for 3 days on MS media + 1% sucrose were transferred to MS plates containing 100 mM NaCl + 1% sucrose. After 4 additional days of growth, hypocotyl elongation on salt media was measured. Plants were always randomly distributed in the growth/treatment chambers.

**Constructs**. To mutate ²⁶YY²⁷ in the CC1 sequence to alanine residues, pENTR_CC1[5] was linearized with ApaI and BamH1. Two mutation PCRs with primers listed in Supplementary Table 1 (primers 10–13) and pENTR_CC1 as a template were performed.

The two overlapping PCR products were purified and subsequently inserted into the linearized vector backbone using the Gibson cloning method[43]. The resulting mutated pENTR-CC1YYtoAA was subsequently inserted into UBNGFPpDEST[43,44] by performing LR reactions of the Gateway cloning system (Invitrogen, USA).

For heterologous protein expression of CC1ΔC223 with a hexa histidine-tag in E. coli, the construct pDEST17CC1ΔC223 was used[5]. To obtain the hexa histidine-tagged, mutated ²⁶YY²⁷ to ²⁶AA²⁷ version of CC1ΔC223, pENTRCC1ΔC223[5] was linearized with ApaI and BamH1. Two mutation PCRs were performed using pENTRCC1ΔC223 as a template and the primers "mut general fw", "YY to AA rev", "YY to AA" fw and "mut general rev" as described above. The two overlapping PCR products were purified and subsequently inserted into the linearized vector backbone using the Gibson cloning method. The resulting pENTR-CC1ΔC223YYAA was subsequently inserted into the Gateway® pDEST™17 Vector (Invitrogen, USA) by performing LR reactions.

For the production of NMR-samples from an ineffective pETM11 His_Sumo_CC1ΔC223 construct the Sumo part was removed and a 3C site was introduced. That construct and two mutants were made by a modified QuickChange protocol, introducing mutation sites via primer amplification (primers 14–17; Supplementary Table 1) of the whole vector by KOD-polymerase (Novagen, Merck, Germany) and subsequent digest of parental vector by DpnI (fast digest, Thermo Fisher Scientific Inc., USA). The setup was transformed into Giga cells (Novagen, Merck, Germany) for plasmid preparation and sequencing (Source Bioscience, UK).

**Representative image acquisition of seedlings**. Representative images of seedlings grown on plates with a diversity of drugs were acquired with a Leica M205 FA or ZEISS Axio Zoom.V16 microscope. Before imaging, the seedlings were transferred to fresh MS plates. The stitching function of Leica LAS X Life Science software was used when seedling did not fit into one image at lowest magnification.

**Heterologous protein expression**. 6xHis-tagged CC1ΔC223 and 6xHis-tagged CC1ΔC223YYAA (using the Gateway® pDEST™17 Vector (Invitrogen, USA) were expressed in Rosetta2 (DE3) pLysS *E. coli* Cells (Merck Millipore, Germany). A starter culture was grown overnight at 28 °C and used to inoculate the main cultures. Cultures were grown at 37 °C until an OD600 ~0.6 was reached and protein expression was induced by addition of isopropyl β-D-1-thiogalactopyranoside (IPTG) at a final concentration of 1 mM. Cells were collected after 3 h by centrifugation, washed in 150 mM NaCl and resuspended in lysis buffer (50 mM Tris-HCl, 200 mM NaCl, 20 mM imidazole, pH 7.4). Cell lysates were prepared by passing the solution through an Microfluidics M-110P Homogenizer (Microfluidics Corp., USA) at 22,000 psi for four times. Cellular debris was spun down, supernatant was collected and filtered through a 0.2 μm filter. Recombinant proteins were purified using Ni Sepharose™ High Performance HisTrap™ HP columns (GE Healthcare Life Sciences, USA) and an ÄKTA pure 25L (GE Healthcare Life Sciences, USA) equipped with a 50 ml Superloop. Buffers for protein purification were prepared as follows: Buffer A: 50 mM Tris-HCl (pH 7.4), 200 mM NaCl; Buffer B: 50 mM Tris-HCl (pH 7.4), 200 mM NaCl, 500 mM Imidazole. Sample application onto the column was performed at 1 ml/min with 92% A and 8% B. The column was washed stepwise at 1.5 ml/min as follows: 20 column volumes (CV) 92% A and 8% B, 6 CV 80% A and 20% B, 6 CV 65% A and 35% B. Final protein sample was eluted at 1.5 ml/min for 10 CV with 100% B in upflow mode. Samples were collected in 2 ml fractions. Fractions enriched with protein were combined and gel filtered to remove imidazole using PD-10 Desalting Columns (GE Healthcare, USA) according to the gravity protocol in the manual. The column was equilibrated with 50 mM Tris-HCl (pH 7.4), 200 mM NaCl. In a final step, proteins were concentrated at 12 °C and 3000 × g using Macrosep® Advance Centrifugal Devices with a 3K cutoff (Pall Corporation, USA). Proteins were snap frozen and stored at −80 °C until further use.

For NMR-scale production of CC1ΔC223 and CC1ΔC223YYAA, pETM11-HisCC1ΔC223 (KanaR) or mutants and the helper plasmid pBAD-σ32 (I54N) (AmpR) were co- transformed into the *Escherichia coli* strain BL21(DE3) star (Thermo Fisher Scientific Inc., USA). σ32-I54N expression was induced at an OD of 0.6 with 0.2% L-arabinose for 2 h on LB medium, followed by the induction of His-CC1ΔC223 with 1 mM IPTG on minimal medium M9 with ¹³C glucose and ¹⁵N NH₄Cl for NMR purposes. Cells were collected by centrifugation after additional 5 h at 37 °C. His-CC1ΔC223 was purified by metal chelate chromatography. The His-tag was cleaved off overnight with 3C/PreScission Protease at 10 °C. The cleaved tag and the protein were separated by gel filtration including the exchange of the buffer to 20 mM Tris-HCl, 200 mM NaCl (pH 7.4). Final samples had a concentration in the range of 1–6 mg/ml (70–500 μM) containing 0.02% sodium azide and protease Inhibitor (cOmplete, EDTA-free; Roche, CH). Extinction coefficients of proteins were determined with ProtParam[45].

**Cross-linking and tandem mass spectrometry**. 6xHis-CC1ΔC223 was purified as described above but 50 mM Tris-HCl (pH 7.4) was exchanged for 50 mM sodium phosphate buffer (pH 7.0). The protein was diluted in the same buffer to a concentration of 1 mg/ml. Tubulin was solubilized to a final concentration of 1 mg/ml in 50 mM sodium phosphate buffer (pH 7.0). 100 μl tubulin and 72 μl 6xHis-CC1ΔC223 were mixed with 28 μl of 50 mM sodium phosphate buffer (pH 7.0). Accordingly, the final concentration of both proteins in the reaction mixture was 40 μM. Cross-linking was initiated by adding 10 μl 1-ethyl-3-(3-dimethylamino-propyl)carbodiimide hydrochloride (EDC) (10 mg/ml in water). The reaction was incubated at RT on a rotating device (600 rpm) for 2 h. Free EDC was removed by gel filtration using PD midiTrap G-25 columns (GE Healthcare, USA) according to the spin protocol in the manual. The column was equilibrated with 50 mM sodium phosphate buffer (pH 7.0) and 200 mM NaCl. 50 μl of the EDC sample was subjected to in solution digestion with trypsin and 75 μl was used for SDS-PAGE. In solution digestion with trypsin ("Trypsin Gold, Mass Spectrometry Grade", Promega, USA) was performed as described by[46]. Deviating from the protocol, no alkylation of cysteines was performed (6xHis-CC1ΔC223 only consists of one cysteine at the C-terminus).

For in gel digestion, the sample was divided into three aliquots and loaded on a gradient protein gel (NuPAGE™ 4–12% Bis-Tris Protein Gels, 1.5 mm, 10-well, Thermo Fisher Scientific). SDS-PAGE was performed as described by ref. [47]. The bands corresponding to the molecular weight of tubulin cross-linked to 6xHis-CC1ΔC223 were excised and combined. In-gel digestion of the cut bands was performed as described by ref. [48]. Again, no alkylation of cysteines was performed. LC/MS/MS analysis of the cross-linked samples was performed on an Orbitrap Fusion™ Lumos™ Tribrid™ Mass Spectrometer (Thermo Fisher Scientific) fitted with a nanoflow HPLC (Ultimate 3000 RSLC, Dionex). The nano-LC system was equipped with an Acclaim Pepmap nano-trap column (Dionex – C18, 100 Å, 75 μm × 2 cm) and an Acclaim Pepmap RSLC analytical column (Dionex – C18, 100 Å, 75 μm × 50 cm). The samples were loaded on the trap column with an isocratic flow of 5 μl/min of 3% CH₃CN containing 0.1% formic acid for 5 min before it was switched in-line with the analytical column. Samples were eluted with 0.1% (v/v) formic acid (A) and 100% CH₃CN/0.1% formic acid (v/v) (B). Solvents were applied with the following gradient: 3% B to 12% B for 1 min, 12% B to 35% B in 43 min, 35% B to 80% B in 2 min. The MS system was operated in positive ion mode at a resolution of 120,000 in full scan mode using data-dependent acquisition (DDA). The MS2 was operated in HCD mode with a resolution of 30,000, AGC

target of 50,000 and Activation Q of 0.25 for ions above 50,000 with a charge state between 3 and 8. Raw data were converted to mgf or mzML format with MSconvert[49] with peak picking enabled. A sequence database comprising the CC1 protein (including N-terminal 6xHis), tubulin beta chain (TBB_PIG: https://www.uniprot.org/uniprot/P02554) and tubulin alpha-1A chain (TBA1A_PIG: https://www.uniprot.org/uniprot/P02550) was employed for all queries. For analysis with StavroX, the four mgf files where combined into a single file with an automated batch file from PROTEIN METRICS.

For the initial identification of cross-linked peptides, four different tools were used: Crux[21], Spectrum Identification Machine (SIM-XL)[20], pLink[19] StavroX[18]. All parameters and settings for the tools can be found in Supplementary Methods section. Spectra identified with Crux were visualized and annotated using mMass (ver. 5.5.0)[50].

Finally, all identified spectra corresponding to cross-linked peptides from the four different tools were manually inspected to ensure assignment of major product ions. The mass spectrometry proteomics data have been deposited to the ProteomeXchange Consortium via the PRIDE[51] partner repository with the dataset identifier PXD009260.

**Microtubule bundling assay and TEM**. A total of 33 μL of 6xHis-CC1ΔC223 (purified as described above in 50 mM Tris-HCl (pH 7.4), 200 mM NaCl) in various concentrations were added on ice to 30 μL of porcine tubulin (Cytoskeleton #240-A) (4 mg/ml) in BRB80 (80 mM PIPES, 1 mM MgCl₂, 1 mM EGTA, pH 6.8 adjusted with KOH) + 1 mM GTP (G8877 Sigma). 10 μL of BRB80 Cushion (80 mM PIPES, 1 mM MgCl₂, 1 mM EGTA, 60% glycerol, pH 6.8 adjusted with KOH) were added in a last step. 6xHis-CC1ΔC223 concentrations were ranging from 1.7 to 5.1 μM. The mixture was incubated at room temperature for 20 min to allow microtubules to polymerize in the presence of 6xHis-CC1ΔC223. As controls, the highest concentration of 6xHis-CC1ΔC223 (5.1 μM) was replaced with an equal amount of BSA (Sigma), or with 50 mM Tris-HCl (pH 7.4), 200 mM NaCl buffer only. Microtubules were pelleted by centrifugation at 30,000×g and room temperature for 30 min, and supernatant was discarded. Microtubules were resuspended in 100 μL BRB80 with 5 μM taxol. 10 μL of sample was applied to copper grids (Gilder) coated with 0.3% formvar (Electron Microscopy Sciences) and absorbed for 30 s, washed once quickly with dH₂O, then stained with 10 μL of 1% aqueous uranyl acetate solution for 60 s, rinsed once quickly with dH₂O, and dried. Samples were imaged on a Philips CM120 BioTWIN transmission electron microscope equipped with a Gatan MultiScan 791 CCD camera and a tungsten filament at an accelerating voltage of 120 kV or a Tecnai G2 Spirit TEM equipped with an FEI Eagle 4K-HS CCD camera and a LaB6 filament at an accelerating voltage of 120 kV. Microtubule bundling was assessed in images by counting the number of microtubules per bundle (degree of bundling) and the percentage of microtubules in images that were incorporated into bundles.

**Gold labeling and TEM of microtubule-associated CC1**. Microtubules were polymerized in the presence of CC1 as described above, but the pelleted microtubules were resuspended in BRB80 without EGTA plus 5 μM taxol. 5 nm gold functionalized with a Ni-NTA group (Nanoprobes #2082) was added to the CC1-microtubule mixture to a final concentration of 1/10 and incubated for 30 min at room temperature. To remove unbound gold, the CC1-microtubule mixture was applied to a 3 k spin column (Pall Naonsep) and centrifuged in 30 s intervals at 600 × g until most of the liquid had flowed through the column, but the microtubules were not dried. The column was rinsed by applying 50 μL BRB80 without EGTA plus 5 μM taxol using the same method. The microtubules in buffer that remained on the column were absorbed onto formvar coated grids, negatively stained, and imaged as described above. Gold distribution along microtubules was quantified using Fiji[52–54].

**Fluorescent labeling of His-CC1ΔC223**. Freshly purified 6xHis-CC1ΔC223 in 50 mM Tris-HCl (pH 7.25), 200 mM NaCl at a concentration of 60 μM was used (see above for purification details). CF488A CF Dye Maleimide (Biotium, Inc., USA) was diluted in anhydrous DMSO to a concentration of 10 mM. The dye was added to the protein sample under constant shaking (650 rpm) in a 10X molar excess. Samples were incubated in the dark/650 rpm at RT for 2 h. The protein was again loaded onto a Ni Sepharose™ High Performance HisTrap™ HP column (GE Healthcare Life Sciences, USA) and washed with 50 mM Tris-HCl (pH 7.4), 200 mM NaCl until the flow through was clear and unbound dye removed. The labeled protein was then eluted with 50 mM Tris-HCl (pH 7.4), 200 mM NaCl, 250 mM imidazole. To remove imidazole and the last traces of unbound dye, the protein was dialyzed in Spectra/Por® Dialysis Tubing (3500 molecular weight cutoff) (Spectrum Laboratories, Inc., USA) overnight against 50 mM Tris-HCl (pH 7.4), 200 mM NaCl. The protein was concentrated at 12 °C and 3,000 × g using a Macrosep® Advance Centrifugal Devices with a 3K cutoff (Pall Corporation, USA). Proteins were snap frozen and stored at −80 °C until further use.

The concentration of the labeled protein was calculated using the following equation:

[conjugate] (mg/mL) = {[A280−(A490 × CF)]/ε} × df; CF = correction factor = 0.1 for CF488A; ε = molar extinction coefficient of 6xHis-CC1ΔC223 in mg/mL = 0.803; df = dilution factor.

The degree of labeling (DOL) was calculated using the following equation:

DOL = (A490 × Mwt × df)/($\varepsilon$ × [conjugate]); Mwt = molecular weight of 6xHis-CC1ΔC223 = 16,122 Da; df = dilution factor; $\varepsilon$ = molar extinction coefficient of CF488A (70,000).

**Microtubule diffusion assay**. Flow cells were made from two glass cover slips and 1 mm wide stripes of double-sided sticky tape (tesa, Germany). General handling was performed as described earlier[26]. Briefly, Rhodamine-labeled tubulin (Cytoskeleton, Inc., USA) was mixed in a 1:20 ration with unlabeled tubulin (Cytoskeleton, Inc., USA). 4 mg/ml of the mixture were polymerized to microtubules, stabilized with taxol and bound onto an anti β-tubulin antibody (1:200 dilution; Sigma Monoclonal Anti-ß-Tubulin #T7816) coated imagining channel as described earlier[26,55]. CF488A-labeled CC1ΔC223 (DOL = 0.11) was mixed 1:4 with unlabeled CC1ΔC223, was then injected into the imaging channel and incubated for 30 s. Unbound protein was washed out with antifade solution[26]. To image the motility of CC1ΔC223 proteins, objective-type total internal reflection fluorescence (TIRF) microscopy was carried out on an inverted AxioObserver equipped with a TIRF-slider system (both from Zeiss, Götttingen, Germany). The slider was fiber-coupled to a 488 nm diode laser (Stradus 488-50, Vortran Laser Technology, Sacramento, CA) and a 532 nm diode-pumped solid-state laser (Cobolt Samba 100 mW, Cobolt AB, Solna, Sweden). The microscope was equipped with a Lumen 200 metal arc lamp (Prior Scientific Instruments, Jena, Germany) to provide fluorescence excitation in epi-illumination. Excitation and detection of fluorescence was achieved using a ×63, NA1.46 alpha-Plan-Apochromat oil immersion objective from Zeiss. Fluorescence filters: Unless otherwise mentioned we used the following filter sets from Semrock (Lake Forest, IL): 1) for CC1ΔC223 proteins, BL HC 482/18, BL HC R488, BL HC 520/35 and 2) for rhodamine-labeled MTs, BL HC 520/35, zt 532 RDC from Chroma (Bellows Falls, VT), HC 585/40. Image acquisition: for capturing fluorescence, we used an electron-multiplied charge-coupled device camera (iXon DV 897E, Andor, Belfast, Northern Ireland). The CC1ΔC223 proteins were imaged using a continuous recording mode (streaming; duration 50 s at 10 frames per s) of the MetaMorph imaging software (Molecular Devices, Sunnyvale, CA).

The recorded movies of the in vitro assays were analysed using the open-source softwares Fiji[52–54] and FIESTA[56]. Movies were background subtracted and frame-registered using standard plugins of Fiji. The intensity of microtubules and CF488A-labeled CC1ΔC223 proteins were measured in average projections of over 500 frames (=50 s) using 10-pixel-wide cross-sects through the microtubule and CC1ΔC223 tracks, respectively. The microtubule numbers in the bundles were counted by starting out measuring the intensity of clearly visible single microtubules. Next we measured the intensity of regions where two single microtubules merged into a defined doublet. Occasionally, we found regions where doublets were joined by another single microtubule. At best, we could differentiate up to five microtubules within one bundle. We then compared the intensity of the different microtubule bundles to the intensity of the CC1ΔC223 proteins accumulated at those regions. The lifetime of CC1ΔC223 proteins on the microtubules was measured by hand using the kymograph tool of FIESTA. Lifetime was measured from the first appearance of a CC1ΔC223 foci until its disappearance. The mean square displacement (MSD) of the CC1ΔC223 proteins was measured by using FIESTA to analyze the diffusion of mobile foci on MTs. After tracking their positions and manually curating the tracks, we calculated the MSD using the corresponding FIESTA plugin.

**Microtubule assembly for NMR**. Porcine brain tubulin was purified as described in ref. [57]. Tubulin was polymerized in microtubule assembly buffer with 100 mM Na-Pipes, pH 6.9, 1 mM EGTA, 1 mM MgSO₄, 1 mM GTP and 1 mM DTT. The concentration of tubulin ranged from 10 to 40 μM and taxol was added in equimolar concentration. The sample was incubated at 37 °C for 45 min and fractionated by ultracentrifugation at 100,000 × g for 1 h. For NMR experiments the pellet was resuspended in a 1:1 mixture of the microtubules assembly buffer and 50 mM Tris-HCl (pH 7.4), 200 mM NaCl. Electron microscopy showed that the microtubules remained stable during the course of the experiments.

**NMR spectroscopy and data analysis**. All experiments were performed at 20 °C at 600 and 750 MHz Bruker Avance spectrometers (Bruker, Karlsruhe, Germany) equipped with cryogenically cooled triple resonance probe heads on samples containing 10% D₂O in the above mentioned buffer ratio. The raw NMR data were collected and processed using the TopSpin software (Bruker, Karlsruhe, Germany). Free $^{15}$N,$^{13}$C-labeled CC1ΔC223 was assigned using three-dimensional HNCACB, HN(CO)CACB and four-dimensional HNCOCA and HNCACO experiments. The two 3D experiments were acquired each with 512 × 44 × 64 complex data points in the direct $F_3$ ($^1$H) and the two indirect $F_2$ ($^{15}$N), $F_1$ ($^{13}$C) dimensions resulting in 51 ms, 22 ms and 6.4 ms of acquisition time, respectively. The 4D data was acquired using non-uniform-sampling with 22% sparse sampling and reconstructed using the MDD routine implemented in the Bruker TopSpin processing software. All spectra were recorded as BEST-type experiments on a sample with a 250 μM concentration[58,59]. Intensity ratios were calculated based on BEST-type HNCA spectra recorded on 150 μM uniformly $^{15}$N–$^{13}$C-labeled samples of CC1ΔC223 and CC1ΔC223YYAA alone and in the presence of 100 μM microtubules. Each experiment was acquired with 16 scans and 512 × 36 × 44 complex

data points, corresponding to an acquisition time of 51 ms in $F_3$ ($^1$H), 22 ms in $F_2$ ($^{15}$N) and 11 ms in $F_1$ ($^{13}$C), respectively. The total measurement time of each experiment was 20 h. The chemical shift analysis for the detection of structure propensity utilized the random-coil chemical shift library ncIDP[28]. The chemical shifts were automatically referenced using the method described by Marsh et al.[60].

The $^{15}$N-$R_2$, $^{15}$N-$R_1$ and CPMG relaxation dispersion measurements were carried out using a $^1$H–$^{15}$N HSQC-based experiment which was recorded as a pseudo 3D with single-FID interleaving and Waltz-16 $^{15}$N-decoupling during acquisition periods. Each 2D plane was comprising 512 × 128 complex data points in the $^1$H (direct, $F_2$) and $^{15}$N (indirect, $F_1$), corresponding to 51 ms and 55 ms of acquisition time, respectively. For the $^{15}$N-$R_2$ experiment, a relaxation-compensated Carr–Purcell–Meiboom–Gill (CPMG) scheme at an effective CPMG field of 550 Hz was applied and relaxation delays were set to 16, 32, 48, 80, 112, 144, 200 ms with 16, 80 and 144 ms recorded twice within one experiment. In the $^{15}$N-$R_1$ experiments the relaxation delays were measured with 80 (2×), 240, 400 (2×), 640, 880 (2×), 1280 and 1600 ms as set delays.

CPMG relaxation dispersion experiments were measured at CPMG fields, $\nu_{CP}$, of 80, 160, 240, 320, 400 (2×), 640, 800, 1000 Hz, with $\nu_{CP} = 1/(2\tau_{CP})$ and $\tau_{CP}$ is the time between 180° $^{15}$N-pulses, which were applied for a constant transverse relaxation time of 100 ms for each CPMG field.

$^{15}$N-$R_2$ and $^{15}$N-$R_1$ experiments were acquired with 16 scans and relaxation dispersion experiments with 64 scans. All experiments employed an interscan delay of 1.3 s resulting in a total measurement time of 17.5 h for the $^{15}$N-$R_2$, 23.5 h for the $^{15}$N-$R_1$ experiment and 26 h for the relaxation dispersion data. To gain $^{15}$N-$R_2$ and $^{15}$N-$R_1$ values, the cross-peaks in the 2D spectra were picked manually and fitted to an exponential function with the software CCPN Analysis. For each fit the error (standard deviation) was estimated using the "covariance" error method. $^{15}$N-$R_{2,\text{eff}}$ values were determined as described in ref. [61]. $^1$H–$^{15}$N NOE values were determined by analysing the ratios of peak intensities in paired NMR spectra with and without 3 s of proton saturation in 64 scans.

All relaxation experiments were recorded on 100 μM uniformly $^{15}$N-labeled CC1ΔC223 before and after adding 25 μM of microtubules. All data were apodized with 90°-shifted sine function and zero-filled to yield appropriate data matrices.

Saturation-transfer difference NMR spectra were recorded using a series of equally spaced 20 ms Gaussian-shaped pulses for a total saturation time of 2 s and an interscan delay of 5 s. On- and off-resonance frequencies were set to −0.5 ppm and 60 ppm, respectively. The measurements were performed on samples containing 1 mM CC1-peptide and 25 μM microtubules in the above mentioned buffer ratio. The 2D $^1$H–$^{13}$C HMQC were recorded on the sample with 1024 × 128 complex data points, corresponding to an acquisition time of 69 ms in the $^1$H (direct, $F_2$) and 14 ms in the $^{13}$C (indirect, $F_1$) dimension. Applying an interscan delay of 1.3 s and 256 scans, the experiment's duration was 25 h.

**Sequence conservation**. Tau and CC1 evolutionary sequence conservation was calculated by CONSURF on approximately 300 closest homologous sequences identified by BLAST for each protein[62,63].

**Analytical ultracentrifugation**. Sedimentation velocity (SV) experiments were performed using a Beckman Optima XL-I analytical ultracentrifuge. Two-channel centerpieces were loaded with 400 μl samples of CC1ΔC223 in dialysis buffer at concentrations ranging from 15 to 60 μM. SV runs were carried out overnight at 35,000 rpm, at 20 °C. Absorbance scans at 280 nm were collected every 5 min. Sedimentation coefficient distributions $c(s)$ were determined using the program Sedfit[64]. The partial-specific volume of CC1ΔC223 was predicted as 0.709 ml/g based on its amino acid sequence using Sednterp[65]. Figures were created with GUSSI (available at http://biophysics.swmed.edu/MBR/software.html).

**Circular dichroism spectroscopy**. 6xHis-CC1ΔC223 was dialyzed against pure water overnight at 4 °C. The sample was spun down at 20,000 × g for 10 min, the supernatant consisting of soluble protein was diluted to 0.1 mg/ml and transferred into a 0.1 mm path length cuvette. The background was recorded with pure water. Circular dichroism (CD) spectra were recorded at room temperature (approx. 22 °C) on a Jasco-715 spectropolarimeter (Jasco Analytical Instruments, USA) which was constantly purged with N₂. A total of 4 spectra were accumulated for each measurement with a response time of 4 s, 1 nm data pitch and a 1 nm bandwidth from 260 to 186 nm. The mean residue weight (mean residue ellipticity) was calculated using the following equation:

θMRW = (θ(λ) × 0.1)/(n × c × d); θ(λ) = recorded spectra in millidegrees, n = number of amino acid residues, c = sample concentration in mol/l, and d = path length of the cuvette in cm.

**Microtubule affinity assay**. Microtubule spin down assays were performed as described in detail before[66]. Briefly, a constant amount of polymerized tubulin (i.e. microtubules) was incubated with increasing concentrations of either 6xHis-CC1ΔC223 or 6xHis-CC1ΔC223YYAA ranging from 0 to 30 μM. The samples were incubated for 30 min at RT and subsequently spun at 30,000 × g for 30 min at RT to pellet microtubules and bound proteins. Supernatant and pellet fractions were subjugated to SDS-PAGE and protein levels in both supernatant and pellet

fractions were analyzed using the Gel-function of Fiji. Final dissociation constant ($K_D$) was estimated by fitting a saturation binding curve onto the data points with GraphPad Prism v8.0 (GraphPad software, Inc., USA).

**Cell wall analysis**. To quantify the cellulose content under salt stress, wild-type and mutant seedlings were grown on control plates (MS media) for 2 days and then transferred to plates containing 150 mM NaCl. After 5 more days of growth, the seed coats were removed and crystalline cellulose was quantified as before[67].

**Live cell-imaging and data processing**. XFP-tagged proteins were imaged with a CSU-W1 Yokogawa spinning disc head fitted to a Nikon Eclipse Ti-E inverted microscope with a CFI PlanApo × 100 N.A. 1.40 oil immersion objective, an EM-CCD ImageEM 1K (C9100-14) (Hamamatsu Photonics, Japan), and a ×1.2 lens between the spinning disc and camera. GFP was imaged using a 488 nm solid-state diode laser and a 525/50 nm emission filter, RFP was detected with a 561 nm solid-state diode laser and a 609/54 nm emission filter. Time lapse images were processed and analyzed with Fiji. Drifts were corrected by using the plugin StackReg or MultiStackReg in cases where two channels were imaged[68]. When the drift of samples could not be corrected in this way, they were excluded from the analysis. Backgrounds were subtracted by the "Subtract Background" tool (rolling ball radius, 30–50 pixels). To quantify CC velocities three frames were averaged by "WalkingAverage" and kymograph analysis was performed with the kymograph tool of FIESTA. Co-localization analysis was performed using the JACoP plugin[69] of Fiji. For the dual-labeled lines GFP-CC1/GFP-CC1YYAA and tdTomato-CesA6 the van Steensel's cross-correlation function (CCF) was determined by shifting one channel in $x$-direction pixel per pixel relative to the corresponding channel and calculating the Pearson coefficient[70]. Correlation maxima at shift = 0 pixels indicates co-localization of both channels for most fluorescent foci.

**Microtubule quantification and particle detection**. Microtubule and particle density at the plasma membrane of cells were determined as described before[5]. Briefly, cortical microtubules were quantified using a Matlab (The MathWorks, Natick, USA) based program, which applied a Sobel edge-detection algorithm at various detection thresholds to raw microscopy images. Detection thresholds were determined manually by choosing a threshold that chose most microtubules and least noise pixels. The cell area was subsequently determined by applying expansion and closing steps to the detected edge-pixels and led to enclosed microtubule regions. Microtubule coverage was then calculated by dividing the microtubule area by the cell area. By using a self-written Fiji macro, particle density at the plasma membrane was determined. The cell area was detected by convoluting a wide Gaussian kernel (sigma = 1.33 mum) into each raw image and applying an auto-mated threshold. Particles were subsequently detected by generating a Laplacian image with FeatureJ (Erik Meijering, Biomedical Imaging Group, EPFL Lausanne) from smoothed images (Gaussian kernel with sigma = 0.2 μm). Peaks were detected using the Find Maxima function with a noise tolerance of 800. Analogously, Golgi were detected (Gaussian kernel with sigma = 0.8 μm; noise tolerance of 120). The number of Golgi was subtracted from the number of total particles. Density was calculated by dividing the resulting particle count by the cell area.

**Quantification of microtubule bundling and dynamics**. Image analysis was done using Fiji. Cell boundaries were detected by convoluting a wide Gaussian kernel (sigma = 10 μm) into each image and thresholding using the Otsu algorithm[71]. All further operations were conducted within these cell boundaries. To detect microtubules and skeletonize the raw images, a Laplacian image was generated using FeatureJ and smoothing of 1.5 (Erik Meijering, Biomedical Imaging Group, EPFL Lausanne). A user defined threshold was then applied for each individual image that covered all microtubules in the image independently of the background. The "Despeckle Filter" of Fiji was then applied. Microtubules were then detected using the "Analyze Particles" function of Fiji with a size from 20 to infinity pixels and the outside of these was cleared. The skeletonized microtubule image was then analyzed using DiameterJ[72] with standard settings. The calculated "Super Pixel" was taken as a measure for mean fiber diameter and therefore bundling of fibers.

Microtubule dynamics were measured as described in detail before[5]. Briefly, time phased image subtraction with a time shift of one frame was applied to identify shrinking and growing microtubule ends. The velocities of these growing and shrinking microtubule ends were finally analyzed with the kymograph evaluation tool of FIESTA.

**Statistical analysis and experimental design**. For statistical analyses, Welch's unpaired $t$-test or two-way ANOVA were performed using GraphPad Prism 8. A $p$-value of < 0.05 was considered as statistically significant. Each statistical method used to calculate $p$-values is defined in the corresponding figure legends. Data analysis (especially for images) was either done automatically, so independently from the investigator, or file names were removed before the analysis. For the measurement of plant size, investigators were not blinded. In this case, data were always collected according to the genotype of plants. Sample size was determined for each experiment based on similar data reported in scientific literature.

**Reporting summary**. Further information on experimental design is available in the Nature Research Reporting Summary linked to this article.

## Data availability
All cross-linking mass spectrometry data are available at PRIDE with the identifier PXD009260. The NMR chemical shift data are deposited in the BMRB database with accession number 27660. Data underlying Fig. 1a, f–h, Fig. 2b, e, 3b–d, Fig. 4c–f, Fig. 5a, c, e–h, Fig. S1a, Fig. S3b, c, e, f, Fig. S4a-g, Fig. 5a-e, g, Fig. S6i-j, Fig. S7d-e, g-h are provided as a Source Data file. Any other data are available from the corresponding authors upon request.

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

## Acknowledgements

Live cell imaging was performed with equipment maintained by the Center for Microscopy and Image Analysis (University of Zurich) and Scientific Center for Optical and Electron Microscopy (ScopeM, ETH Zurich). We thank Martina Leidert and Natalja Erdmann for help with protein expression and purification and Ines Kretschmar for synthesizing the peptides. We thank Dmytro Puchkov for performing the MT quality control by EM and Tim Scholz (Hannover Medical School) for assistance with the microtubule purification protocol. We also thank the Diez group, especially Felix Ruhnow, (B CUBE Center for Molecular Bioengineering, Dresden) for assistance with microtubule imaging and in vitro assays. The helper plasmid pBAD-σ32 (I54N) was a gift from Jeffery Kelly[73] (Addgene plasmid # 59982), pETM11 was kindly supplied by EMBL Protein Production facility (Heidelberg Germany). We thank the Biological Optical Microscopy Platform, the Melbourne Advanced Microscopy Facility, and the Mass Spectrometry and Proteomics Facility (School of Biosciences and Bio21) at the University of Melbourne. S.P. was supported by an ARC Discovery grant (DP190101941), a Hermon-Slade Grant (Persson HSF 15/4) and a Future Fellowship grant (FT160100218). J.L.H. was supported by an ARC Future Fellowship (FT130101165). H.E.M. was supported by an ARC DECRA (DE170100054). C.S.R. and C.K. were supported by ETHZ and a SNF grant (2-77212-15). R.S. received Computational Biology Research Initiative and Early Career Research Grants from the University of Melbourne. C.K. was supported by a Peter und Traudl Engelhorn-Stiftung fellowship.

## Author contributions

C.K., A.W., R.S., H.E.M., A.D., B.J.v.R., E.R.L., J.H., C.S.R., H.O. and S.P. designed the research. C.K., A.W., R.S., H.E.M., A.D., M.L., G.A.K., N.C., P.S., F.S. and K.F. performed the research. C.K., A.W., R.S., H.E.M., W.G.S., J.H. and S.P. analyzed data. C.K., A.W., C.S.R., H.O. and S.P. wrote the article. C.K. and A.W. share equal first authorship.

## Additional information

**Competing interests:** The authors declare no competing interests.

