## [Peer Review File · Nature Communications]

Reviewers' comments:

Reviewer #1 (Remarks to the Author):

The manuscript describes studies focusing on a plant protein that binds to microtubules. Using microscopy the authors demonstrate that this protein binds to and diffuses along microtubules in a similar manner as the well-characterized MAP tau. Using NMR spectroscopy the authors further identify four regions that preferentially interact with microtubule. Based on these data they then introduce an artificial mutation into one of these sites and demonstrate that the mutant protein has a perturbed interaction with microtubules in vitro and in cells. The manuscript is very well written and nicely illustrated.

Major points:

The authors stress in their work that the mechanism of binding to microtubules of this plant protein is very similar to the well-studied tau protein. This interpretation is nicely supported by the data. At the same time, this finding questions whether the study is appropriate for a high impact journal such as Nat Comm.

Along the same line, the authors hypothesise in the discussion section that there are also some differences between this plant protein and tau (as suggested by differences in localisation). However, the current study fails short to provide any insights into these differences.

Another weak point is that the authors summarise their findings in a figure illustrating the binding mode of the plant protein to microtubules. But this looks very similar to cartoon representations previously published for tau binding to microtubules. Although this might be reasonable based on the obtained data, it stresses a further weakness of the manuscript: we do not learn anything new about how MAPs bind to microtubules.

Technical issues:

- Fig. 3D is very small and it is impossible to see the secondary chemical shifts of different nuclei
- CSI in Fig. 3D: The CSI makes only sense for folded proteins. For IDPs, a number of other tools have been developed to estimate secondary structure propensities.
- Fig. S3B-D: Again too small to see residue-specific variations. In addition, the hetNOE values appear to be pretty large for a disordered protein.
- Fig. S4D: In order to better judge the contribution of MT-binding to R2 rates the R2 rates in the absence of Mts should be included.

Reviewer #2 (Remarks to the Author):

Using crosslinking MS to reveal binding sites is general a good idea and well-suited to this manuscript. However, from the technical point of view, I have some doubts on the quality of the crosslinks. For instance, the authors used four different software to cumulatively identify five crosslinks. I would expect the confident crosslinks should be identified by all four software.

1. I would like to authors to provide the number of intra-protein links. This should be a relatively high number which ensures the correct use of the method.
2. I think the authors probably have tried Lys-Lys crosslinkers since they are much more commonly used than EDC. I would like to know if the authors find any crosslinks from Lys crosslinkers. And if not, what could be the potential reasons?
3. It would be nice if the authors provide annotated spectra of the five identified crosslinks in the supplementary. They might have uploaded onto the Pride but I don't yet have the reviewer's access to check on them. If some of the crosslinks are only identified by one software, they at least should have a high quality spectra.

Reviewer #3 (Remarks to the Author):

The manuscript entitled "A molecular mechanism for salt stress-induced microtubule array formation in Arabidopsis" by Kesten et al. is a very interesting work. This is a subsequent work to a previous report by the same research group, which had demonstrated an important role of CC proteins to support microtubule and cellulose synthase activity during salt stress (Endler et al., 2015, Cell). In this study, they further analyzed the molecular details of how CC1 interacts with microtubules. They found that N-terminus of CC1 (CC1 Δ C223), which is critical to CC1's function during stress, is intrinsically unstructured and links microtubules through four conserved hydrophobic binding motifs. NMR analyses also revealed two neighboring Tyrosine residues in the CC1 Δ C223 are crucial for the microtubule interaction, as well as for sustaining microtubule array organization and cellulose synthesis during salt stress. This study reports the first Tau-related MAP in plants, and analyzed and discussed conserved structure and MAP function of CC1, as well as its plant specific features. This work is important for better understating this class of MAPs functions, and is of great interests for broad readers.

The authors have demonstrated that CC1 is important for microtubule reorganization and sustained cellulose synthesis during salt stress, I just wonder does salt stress induced any change in CC1 transcriptional expression or protein stability, in addition to previously reported relocation from PM to smaCC-related compartments? Also, in normal growth condition, would PM-localized CC1 (connected to the CSC as well) regulate cortical microtubules?

We thank the reviewers for their comments. All changes made in the manuscript are highlighted in green.

Reviewers' comments:

Reviewer #1 (Remarks to the Author):

The manuscript describes studies focusing on a plant protein that binds to microtubules. Using microscopy the authors demonstrate that this protein binds to and diffuses along microtubules in a similar manner as the well-characterized MAP tau. Using NMR spectroscopy the authors further identify four regions that preferentially interact with microtubule. Based on these data they then introduce an artificial mutation into one of these sites and demonstrate that the mutant protein has a perturbed interaction with microtubules in vitro and in cells. The manuscript is very well written and nicely illustrated.

Major points:

The authors stress in their work that the mechanism of binding to microtubules of this plant protein is very similar to the well-studied tau protein. This interpretation is nicely supported by the data. At the same time, this finding questions whether the study is appropriate for a high impact journal such as Nat Comm.

Response: The reviewer is correct that the CC1 and Tau appears to be mechanistically linked. However, our work does certainly provide new insights into how microtubule-associated proteins work, as:

- 1. CC1 is a transmembrane protein and the microtubule-associated mechanism by which the protein functions is thus in a different cellular context as that of Tau.**
- 2. We show that two conserved tyrosine residues in the first microtubule-binding site of CC1 are important for the protein to engage with microtubules.**
- 3. Our data outline a molecular mechanism for how plants can re-assemble microtubules and sustain cellulose synthesis during exposure to salt. We argue that this is a biological question that is very different from those that Tau are associated with.**
- 4. Finally, our results might provide for approaches to develop salt tolerant plants, which is of major importance in times of environmental and climate change.**

Along the same line, the authors hypothesise in the discussion section that there are also some differences between this plant protein and tau (as suggested by differences in localisation). However, the current study fails short to provide any insights into these differences.

Response:

The reviewer is right in stating that the differences between Tau and CC1 were not addressed in detail. We re-arranged the discussion into three paragraphs (CC1 function and features, similarities of the microtubule binding mechanism to Tau, differences to

Tau) and outlined the clear differences between CC1 and Tau in the last paragraph. Additionally, we moved previous figure S4h to Figure 6 (now Fig. 6a) to emphasize the differences between Tau and CC1 in domain architecture.

In our study we outline:

- how CC1 regulates bundling and dynamics of microtubules
- that CC1 links microtubules at points that are evenly distributed between neighboring microtubules
- that CC1 holds four microtubule-binding motifs and that these engage with microtubules in a highly dynamic manner
- that the first motif contains two tyrosine residues that are important for the microtubule-binding
- we then show that these residues are important for microtubule-based guidance of the cellulose synthesizing protein complex
- and finally that they are needed to sustain salt tolerance in plants, which corroborates our *in vitro* results

This is arguably a comprehensive analysis of CC1 and its molecular function (see also reviewer 3's comments). We hope that our paper will productively impact future research by sparking interesting discussions between otherwise unrelated research fields.

Another weak point is that the authors summarise their findings in a figure illustrating the binding mode of the plant protein to microtubules. But this looks very similar to cartoon representations previously published for tau binding to microtubules. Although this might be reasonable based on the obtained data, it stresses a further weakness of the manuscript: we do not learn anything new about how MAPs bind to microtubules.

Response: We agree with the reviewer regarding the figure and therefore made major figure revisions (please also see our answer to the previous comment). The figure now showcases the differences in the domain structure of CC1 and Tau, but highlights the similarities in the microtubule-binding motifs in terms of amino acid sequence (Fig. 6a). The lower part of the figure (Fig. 6b) now puts CC1 in context of its biological function, i.e. the protein is in close association with the cellulose synthase complex from where it impacts microtubule organization and stability. All components of the figure are depicted to scale, to show a realistic outline of the interaction between CC1 and microtubules. We thus feel that the figure is a useful summary of our obtained results and that it highlights both similarities and differences between Tau and CC1. Please see the response to point 1 above, and also note reviewer 3's comments.

Technical issues:

-Fig. 3D is very small and it is impossible to see the secondary chemical shifts of different nuclei

Response: For more clarity, we enlarged the secondary chemical shifts and moved them to the Supplementary Figure S3B-D.

-CSI in Fig. 3D: The CSI makes only sense for folded proteins. For IDPs, a number of other tools have been developed to estimate secondary structure propensities.

Response: We thank the reviewer for the comment. To provide a more appropriate estimation for secondary structure propensity, we chose the neighbour-corrected Structural Propensity Calculator (ncSPC) tool using the ncIDP reference library for the data analysis instead of CSI. The ncIDP library was specifically compiled for disordered proteins and the ncSPC tool employs a refined version of the SSP score, which has been shown to detect meaningful structural propensities in IDPs. The results are shown in Figure 3d and S3b-d, but do not vary drastically from our previous approach. We have changed the text of the manuscript to describe the analysis. The data can be accessed at the BMRB data base (<http://deposit.bmrb.wisc.edu/bmrb-adit/access.html>) with the restart ID 2018-10-23.deposit.bmrb.wisc.edu.80.56513581.

-Fig. S3B-D: Again too small to see residue-specific variations. In addition, the hetNOE values appear to be pretty large for a disordered protein.

Response: We increased the size of the figures S3B-D (in the revised manuscript: S3 e-g) and added the protein sequence to improve the visibility of residue-specific variations. The hetNOE values are comparable to published data of IDPs of similar size. For example, ¹⁵N-¹H NOE data of the IDP alpha-synuclein shows values between 0 and 0.5 like our measurements on CC1ΔC223 (Theillet, F.-X. et al. Structural disorder of monomeric α-synuclein persists in mammalian cells. Nature 530, 45–50 (2016)).

-Fig. S4D: In order to better judge the contribution of MT-binding to R2 rates the R2 rates in the absence of Mts should be included.

Response: The R₂ rates in the absence of MTs were shown in Figure S3C of the submitted manuscript (S3f in this revised manuscript). To improve clarity, we now show the R₂ values before and after addition of MTs in Figure S4c and further illustrate how the change in the transverse relaxation rate ΔR₂ is generated.

Reviewer #2 (Remarks to the Author):

Using crosslinking MS to reveal binding sites is general a good idea and well-suited to this manuscript. However, from the technical point of view, I have some doubts on the quality of the crosslinks. For instance, the authors used four different software to cumulatively identify five crosslinks. I would expect the confident crosslinks should be identified by all four software.

Response: The reviewer makes a valid point regarding confidence in assigning cross-links in MS. Software used to identify cross-link events from MS/MS data have been developed by various independent research groups, each has its own algorithm or process and each has reported advantages and disadvantages. As a result, we sought to employ several available software packages and apply reported cut-offs to produce a collection of potential inter-cross-link candidates. Each of these reported cross-links was manually inspected to ensure only high-confidence assignments were reported in this manuscript (Table S1). The data are overall consistent and not cumulative, for example all four software packages identified the K124-E111, two software packages identified the K124-E158 cross-link, while other reported cross-links are unique for a given software package. These results are consistent when comparisons are conducted between different proteomic identification software packages (e.g. Mascot vs Sequest), where a large proportion of identifications are shared but then each package, which employs a unique algorithm, matches a unique set of peptides/proteins. We have now added annotated spectra for each of the reported cross-links (Fig S7).

1. I would like to authors to provide the number of intra-protein links. This should be a relatively high number which ensures the correct use of the method.

Response: Indeed, there were many more reported intra-protein cross-links identified by all four software packages. Please see the table below:

	in gel		in sol		
	intra	inter	intra	inter	
SIMXL	53	13	72	36	
Stavrox	80	26	172	12	
pLINK	40	6	130	18	
crux	31	15	3768	34	
Sum (cummulative)	204	60	4142	100	

A total of 4,346 intra-protein cross-links were identified by the four software packages compared to 160 inter-proteins cross-links (pre-manual curation). These numbers are now reported in the figure legend of Table S1 to provide more context and highlight the extent of high-confidence manual curation that was undertaken to arrive at the data in Table S1. Another point highlighting the validity of our analysis is the large number of

identified intra-protein cross-links in the in solution digestion approach (all cross-linked peptide species are present during mass spec analysis) in comparison to the in gel digestion approach (in which we cut the specific cross-linked band of the tubulin dimer + CC1ΔC223).

2. I think the authors probably have tried Lys-Lys crosslinkers since they are much more commonly used than EDC. I would like to know if the authors find any crosslinks from Lys crosslinkers. And if not, what could be the potential reasons?

Response: Choosing EDC over Lys-Lys cross-linkers originated from the following reasoning: The surface of the tubulin dimer is negatively charged (pI around 4.8, please see figure below). Thus, Glu and Asp are the predominant potential cross-linking sites rather than Lys. CC1ΔC223 on the other hand is very positively charged (pI around 8.8). Since EDC links Lys to Asp/Glu, it was a reasonable choice for the cross-linking experiments providing many opportunities for intermolecular cross-links on all three proteins (CC1ΔC223 and tubulins). Conversely, using a Lys only cross-linker may have resulted in a loss of possible cross-link sites at the tubulin surface. Indeed, Table S1 reveals that all identified cross-links are Lys on CC1ΔC223 and Asp/Glu on the tubulin surface. We have now added the reasoning behind our cross-linker choice to the manuscript.

Electrostatic potential mapped onto the tubulin dimer: (a,d) show the side view and the top view of the electrostatic potential distribution on the surface of the tubulin dimer, respectively (filled space, red = negative, blue = positive). Figure and legend were taken from:

Li, L., Alper, J. & Alexov, E. Cytoplasmic dynein binding, run length, and velocity are guided by long-range electrostatic interactions. *Sci. Rep.* 6, 31523 (2016).

3. It would be nice if the authors provide annotated spectra of the five identified crosslinks in the supplementary. They might have uploaded onto the Pride but I don't yet have the reviewer's access to check on them. If some of the cross-links are only identified by one software, they at least should have a high quality spectra.

Response: Unique, annotated spectra for the five reported cross-links are now provided in Fig S7. The raw data at PRIDE can be accessed using the following information (we are sorry that this got lost during the submission process):

Project Name: Cross-linking of the microtubule binding N-terminal domain of the companion of cellulose synthase 1 protein (CC1ΔC223) from Arabidopsis thaliana with tubulin

Project accession: PXD009260

Username: reviewer41181@ebi.ac.uk

Password: usUkHkZE

Reviewer #3 (Remarks to the Author):

The manuscript entitled “A molecular mechanism for salt stress-induced microtubule array formation in Arabidopsis” by Kesten et al. is a very interesting work. This is a subsequent work to a previous report by the same research group, which had demonstrated an important role of CC proteins to support microtubule and cellulose synthase activity during salt stress (Endler et al., 2015, Cell). In this study, they further analyzed the molecular details of how CC1 interacts with microtubules. They found that N-terminus of CC1 (CC1 Δ C223), which is critical to CC1’s function during stress, is intrinsically unstructured and links microtubules through four conserved hydrophobic binding motifs. NMR analyses also revealed two neighboring Tyrosine residues in the CC1 Δ C223 are crucial for the microtubule interaction, as well as for sustaining microtubule array organization and cellulose synthesis during salt stress. This study reports the first Tau-related MAP in plants, and analyzed and discussed conserved structure and MAP function of CC1, as well as its plant specific features. This work is important for better understating this class of MAPs functions, and is of great interests for broad readers.

The authors have demonstrated that CC1 is important for microtubule reorganization and sustained cellulose synthesis during salt stress, I just wonder does salt stress induced any change in CC1 transcriptional expression or protein stability, in addition to previously reported relocation from PM to smaCC-related compartments? Also, in normal growth condition, would PM-localized CC1 (connected to the CSC as well) regulate cortical microtubules?

Response:

CC1 gene expression is highly upregulated upon salt stress but not under osmotic stress which fits nicely with our previous report (see figure below; see also Endler et al., 2015).

Image taken from Arabidopsis eFP Browser (<http://bar.utoronto.ca/efp/cgi-bin/efpWeb.cgi>).

The image summarizes results of:

Kilian, J. et al. The AtGenExpress global stress expression data set: protocols, evaluation and model data analysis of UV-B light, drought and cold stress responses. Plant J. 50, 347–363 (2007).

Protein stability is not easy to assess since it is particularly difficult to pull down intact CSC complexes of which CC1 is part of. Nevertheless, the CC1 is clearly still present with internalized CESAs (in smaCCs/MASCs) during early phases of salt treatment and are then returning with the CESAs to the plasma membrane during later stages of treatment (see figures in current manuscript and in Endler et al., 2015). We therefore do not think that the stability would be much impacted by the stress.

The CC1 might be involved in some aspects of microtubule regulation also during normal conditions. For example, the artificial YYAA mutation in the CC1 caused reduced co-localization of CesAs and microtubules. This indicates that the CC1 does impact the connection between the CESAs and microtubules also during non-stressed conditions. Beyond this, we did not observe any major defects in microtubule regulation; however, it may be important to note that our normal conditions (i.e. growth cabinets/rooms) may be quite different from the environments a plant normally is exposed to and under those conditions the CC proteins might contribute to microtubule regulation much more broadly.

REVIEWERS' COMMENTS:

Reviewer #1 (Remarks to the Author):

The authors have addressed my comments and improved the manuscript.

Reviewer #2 (Remarks to the Author):

The authors have satisfactorily addressed my concerns.